# Identification of Meibomian gland stem cell populations and mechanisms of aging

Xuming Zhu [1,2,3,20], Mingang Xu[1,2,3,20], Celine Portal [4,5,18], Yvonne Lin [4,5,19], Alyssa Ferdinand[1,2,3], Tien Peng [6], Edward E. Morrisey[7], Andrzej A. Dlugosz [8,9,10], Joseph M. Castellano [1,2,11,12], Vivian Lee [13,14], John T. Seykora[14], Sunny Y. Wong[8,10], Carlo Iomini [4,5] ✉ & Sarah E. Millar [1,2,3,15,16,17] ✉

Meibomian glands secrete lipid-rich meibum, which prevents tear evaporation. Aging-related Meibomian gland shrinkage may result in part from stem cell exhaustion and is associated with evaporative dry eye disease, a common condition lacking effective treatment. The identities and niche of Meibomian gland stem cells and the signals controlling their activity are poorly defined. Using snRNA-seq, in vivo lineage tracing, ex vivo live imaging, and genetic studies in mice, we identify markers for stem cell populations that maintain distinct regions of the gland and uncover Hedgehog (Hh) signaling as a key regulator of stem cell proliferation. Consistent with this, we show that human Meibomian gland carcinoma exhibits increased Hh signaling. Aged glands display decreased Hh and EGF signaling, deficient innervation, and loss of collagen I in niche fibroblasts, indicating that alterations in both glandular epithelial cells and their surrounding microenvironment contribute to age-related degeneration. These findings suggest new approaches to treat aging-associated Meibomian gland loss.

Meibomian glands (MGs) are specialized holocrine sebaceous glands (SGs) embedded in the tarsal plate of the upper and lower eyelids. These glands secrete lipid-rich meibum, which protects the eye surface by preventing tear film evaporation. In aging, MGs decrease in size and/or display reduced numbers of acini in humans and in mice, which provide a genetically manipulable model system for delineating molecular mechanisms controlling MG function. MG atrophy is associated with evaporative dry eye disease, a common condition in the

[1]Black Family Stem Cell Institute, Icahn School of Medicine at Mount Sinai, New York, NY 10029, USA. [2]Institute for Regenerative Medicine, Icahn School of Medicine at Mount Sinai, New York, NY 10029, USA. [3]Department of Cell, Developmental and Regenerative Biology, Icahn School of Medicine at Mount Sinai, New York, NY 10029, USA. [4]Department of Ophthalmology, Wilmer Eye Institute, Johns Hopkins University School of Medicine, Baltimore, MD 21231, USA. [5]Department of Cell Biology, Johns Hopkins University School of Medicine, Baltimore, MD 21231, USA. [6]Department of Medicine, University of California San Francisco, San Francisco, CA 94143, USA. [7]Department of Medicine, Perelman School of Medicine at the University of Pennsylvania, Philadelphia, PA 19104, USA. [8]Department of Dermatology, University of Michigan Medical School, Ann Arbor, MI 48109, USA. [9]Rogel Cancer Center, University of Michigan Medical School, Ann Arbor, MI 48109, USA. [10]Department of Cell and Developmental Biology, University of Michigan Medical School, Ann Arbor, MI 48109, USA. [11]Nash Family Department of Neuroscience, Department of Neurology, Friedman Brain Institute, Icahn School of Medicine at Mount Sinai, New York, NY 10029, USA. [12]Ronald M. Loeb Center for Alzheimer's Disease, Icahn School of Medicine at Mount Sinai, New York, NY 10029, USA. [13]Department of Ophthalmology, Perelman School of Medicine at the University of Pennsylvania, Philadelphia, PA 19104, USA. [14]Department of Dermatology, Perelman School of Medicine at the University of Pennsylvania, Philadelphia, PA 19104, USA. [15]Department of Oncological Sciences, Icahn School of Medicine at Mount Sinai, New York, NY 10029, USA. [16]Tisch Cancer Institute, Icahn School of Medicine at Mount Sinai, New York, NY 10029, USA. [17]Department of Dermatology, Icahn School of Medicine at Mount Sinai, New York, NY 10029, USA. [18]Present address: Sorbonne Université, INSERM, CNRS, Institut de la Vision, 17 rue Moreau, F-75012 Paris, France. [19]Present address: Graduate School of Biomedical Sciences, Icahn School of Medicine at Mount Sinai, New York, NY 10029, USA. [20]These authors contributed equally: Xuming Zhu, Mingang Xu. ✉e-mail: ciomini1@jhmi.edu; sarah.millar@mssm.edu

aged population that can result in severe vision loss and currently has no effective therapy[1–3].

Each MG consists of multiple acini, which are connected to a central duct through short ductules. The MG central duct is a stratified squamous epithelial structure that includes basal and suprabasal layers[4,5]. In MG acini, basal cells proliferate toward the center of acinus, forming meibocytes which undergo sequential differentiation and accumulate lipid content. Upon reaching the center of acinus, fully differentiated meibocytes disintegrate and release meibum which reaches the ocular surface through an orifice in the eyelid margin via the ductule and central duct[4,5]. MGs develop independently of hair follicles (HFs) but share similarities with HF-associated SGs. Both types of glands secrete lipids in a holocrine manner and require constant replenishment of lipid-releasing cells from proliferating basal cells[4].

In addition to other proposed mechanisms, including duct hyperkeratinization, immune cell infiltration, and reduced acinar basal cell proliferation[6], MG dropout is thought to involve decreased activity of MG epithelial stem cells that normally maintain MGs by both self-renewing and yielding differentiating progeny[7–11]. However, the identities of MG stem cells, the nature of their supporting niche environment, and the mechanisms that control their functions in homeostasis and aging are poorly understood[12]. By contrast, single-cell analyses and lineage tracing studies in mice have shown that *Lrig1*[+] and *Lgr6*[+] stem cells from the HF isthmus, which connects to the opening of the SG, contribute to homeostasis of the attached SGs[13–16]. Recent evidence suggests that the MG duct and acinus are separately maintained by distinct unipotent KRT14[+] stem cells[12]; however, KRT14 is ubiquitously expressed in the MG[17]. MG ductules have also been proposed to harbor stem cells, based on the presence of label-retaining cells (LRCs) in ductules[12], but evidence that these LRCs directly contribute to MG homeostasis is lacking. Lineage tracing with a constitutive *Krox20/Egr2-Cre* mouse line shows that EGR2[+] cells give rise to the MG during development. However, lineage tracing experiments with an inducible *Egr2-Cre* line to determine whether EGR2[+] cells self-renew and give rise to differentiating progeny in adult MGs have not been performed. Thus, specific markers for adult MG stem cells have yet to be identified.

Stem cell activity is tightly regulated by intercellular signaling pathways[18,19]. These include the Hedgehog (Hh) pathway, which controls HF stem cell activity[20], regulates SG development[21], and promotes proliferation of undifferentiated human sebocytes in vitro[22]. Components of the Hh pathway are expressed in MGs[23], and Hh signaling controls the proliferation and differentiation of rat MG epithelial cells in vitro[24]; however, it is unknown whether Hh signaling regulates adult MG homeostasis and MG stem cell activity in vivo.

To begin to address these questions, we dissected tarsal plates from young and aged mice and performed single-nucleus RNA sequencing (snRNA-seq) to reveal the cell heterogeneity of adult mouse MGs and the changes in MG epithelial cells and the surrounding tissue niche that take place during aging. We chose to employ snRNA-seq rather than scRNA-seq because lipid-rich differentiated MG acinar cells are fragile and cannot easily be collected by centrifugation[16,25]; by contrast nuclei from lipid-rich cells can be pelleted easily allowing us to capture the full range of MG cells including differentiated meibocytes. We used lineage tracing and ex vivo live imaging assays to test the contributions of cells expressing putative stem cell markers to MG homeostasis. These data identified multiple stem cell populations that can self-renew and produce differentiated progeny to maintain the MG duct and acinus. We found that loss of the Hh receptor Smoothened (SMO) in MG epithelial cells caused decreased proliferation and MG dropout, while forced expression of an activated form of the Hh pathway effector GLI2 (GLI2ΔN) promoted expansion of MG stem cells at the expense of differentiated acini. GLI2 and other stem cell markers were expressed in human MG and showed elevated levels in MG carcinoma (MGC), suggesting that MGC involves expansion of MG stem cells. Aged versus young mouse MGs displayed fewer acinar basal cells

and surrounding dermal cells expressing downstream Hh pathway genes. In parallel, aged MG exhibited lower levels of EGF signaling, decreased MG innervation, and dermal fibroblast dysfunction, identifying additional mechanisms that could contribute to MG degeneration in aging. In summary, our data uncover stem cell populations and molecular mechanisms that sustain the adult MG and are altered in aging and in human disease.

## Results

### snRNA-seq identifies multiple distinct populations of MG epithelial cells

The eyelid is a complex environment, containing epidermal, HF, and conjunctival cells closely adjacent to the MGs, making it challenging to isolate pure MG cells or to identify MG contributions in bulk RNA-seq analyses. To analyze transcription in MG subpopulations, and to overcome difficulties in isolating cytoplasmic mRNAs from lipid-rich acinar cells, we utilized snRNA-seq, which allows for identification of cell subpopulations within a complex mix. Nuclei were analyzed from the pooled tarsal plates of four 8-week-old and four 21-month-old male mice using two replicates for each age (16 mice analyzed in total). UMAP analysis identified 34 clusters of tarsal plate cell populations (Fig. 1a; Supplementary Fig. S1a) and confirmed similar cluster distributions between the four datasets and two conditions (Supplementary Fig. S1b–d). Cell populations were annotated based on their expression of known signature genes (Supplementary Fig. S1a). MG ductal basal cells (cluster 3) were marked by expression of *Col17a1* and *Ki67* (Supplementary Fig. S1a, e, f); ductal suprabasal cells (cluster 8) by high levels of *Krt17* and *Egr2* (Supplementary Fig. S1a, f, m–o); orifice cells (cluster 11) by *Krt10*, *Flg* and *Lor* (Supplementary Fig. S1a, g, h); acinar basal cells (cluster 16) by expression of *Pparg*, *Ki67*, and *Slc1a3* (Supplementary Fig. S1a, i, j); ductular cells (cluster 17) by moderate expression of *Krt17* and *Lrig1* (Supplementary Fig. S1m–o), and low levels of *Slc1a3* and *Pparg* (Supplementary Fig. S1a, j); differentiating meibocytes (cluster 24) by expression of *Fasn* and high levels of *Pparg* and *Scd3* (Supplementary Fig. S1a, k, p); and differentiated meibocytes (cluster 18) by expression of *Scd4* and *Scd3*, along with lack of *Fasn* expression (Supplementary Fig. S1a, k, l, p). While *Dnase2* has been described as a marker for differentiated meibocytes in human MGs[26], we found that it is also expressed in other cell sub-populations in murine MGs (Supplementary Fig. S1p); this could be due to species-specific differences. Non-MG populations included conjunctival cells (clusters 0–2, 4–7, 9, 10, 12, 13, 25, 28, 30, 31), HF epithelial cells (clusters 14, 19, 26), HF dermal papilla cells (cluster 32), eyelid dermal cells (cluster 15 and 23), immune cells (cluster 20), endothelial cells (clusters 21 and 29), melanocytes (cluster 33), and muscle cells (clusters 22 and 27).

To verify the accuracy of these annotations, we conducted spatial transcriptomic analysis and integrated the outcomes of the two assays. These analyses showed that the annotation of MG cell populations based on snRNA-seq data was congruent with their anatomical positions within the tarsal plate (Supplementary Fig. S1q). The spatial locations and marker genes for major populations of MG and surrounding stromal cells are summarized schematically in Fig. 1b.

Comparison of these data with published single-cell analysis of the HF-associated SG[16] revealed many commonalities, including expression of low levels of *Pparg* in acinar basal cells. A key difference was that genes associated with neural guidance, such as *Slit3* and *Robo2*, were expressed at relatively higher levels in MG acinar basal and ductular cells compared with HF-associated SG basal cells (Supplementary Fig. S1r, s). In line with this, MGs, but not HF-associated SGs, are directly innervated[27–30].

We used Velocity and pseudotime analyses to analyze the snRNA-seq data for the differentiation trajectories of MG cell populations. These analyses predicted that ductal basal cells give rise to ductal suprabasal cells, while acinar basal cells produce progeny that form differentiating acinar cells, which further differentiate into differentiated acinar cells (Fig. 1c, d). These predictions are in line with those

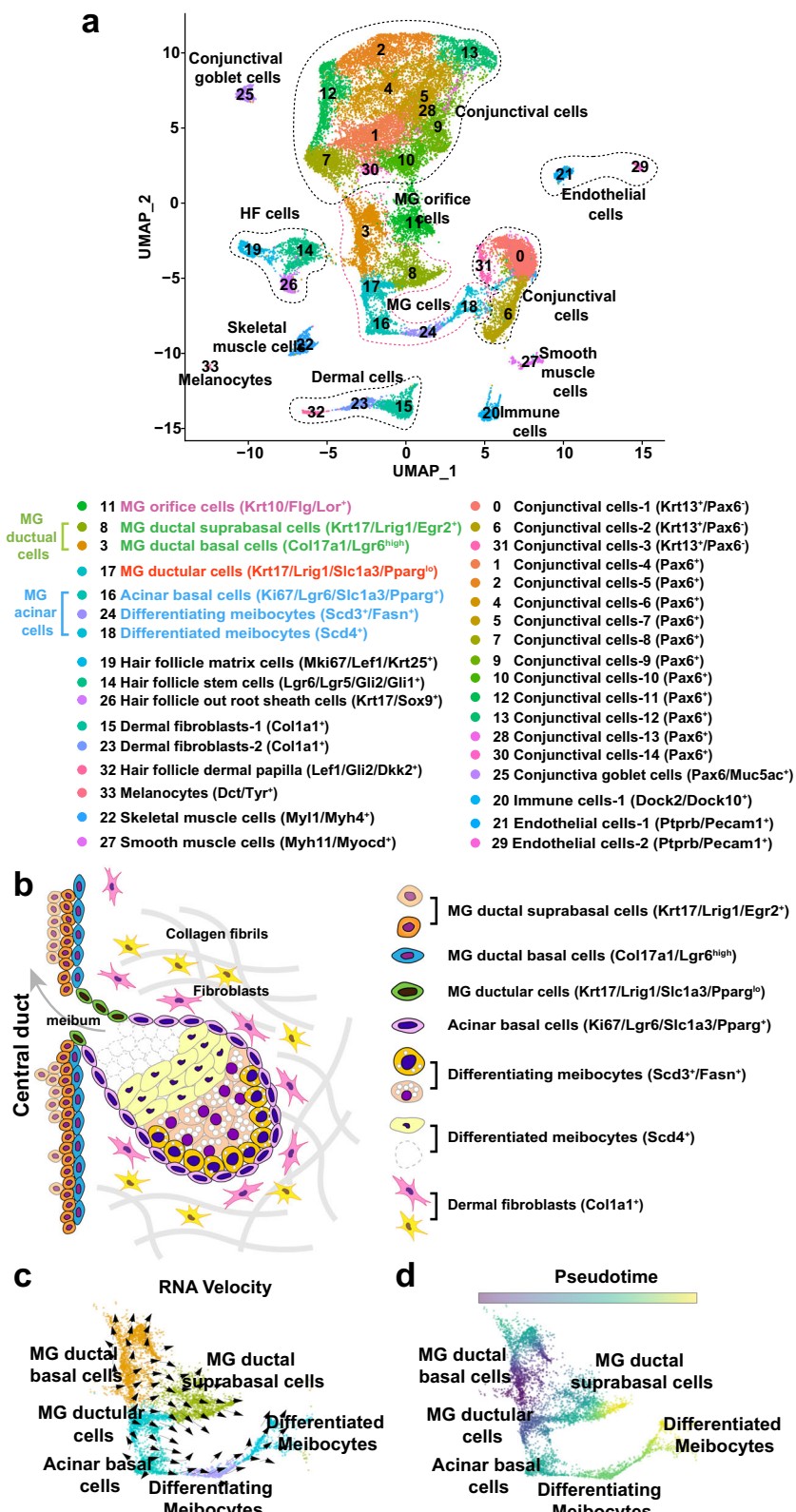

**Fig. 1 | snRNA-seq identifies tarsal plate and MG cell subpopulations. a** UMAP plot of snRNA-seq data from pooled tarsal plate samples from four 8-week-old and four 21-month-old male C57BL/6 J mice, respectively, with two replicates for each age. **b** Schematic diagram of MG structure indicating the spatial locations of MG subpopulations and the stromal environment. **c**, **d** RNA Velocity (**c**) and pseudo-time analyses (**d**) predictions of the differentiation trajectories of MG cellular subpopulations. RNA Velocity (**c**) predicts that ductular cells can differentiate towards either duct or acinus. See also Supplementary Fig. S1.

from recent studies that used scRNA-seq to analyze 6-month-old MGs[25], and HF-associated SGs[16]. Interestingly, however, our analysis also predicted that ductular cells can contribute to both ductal and acinar basal cell populations (Fig. 1c), suggesting that the ductules contain mixtures of stem cells fated to become either duct or acinus, and/or harbor bi-potential stem cells[31].

## MG acinar basal cells and ductular cells express HF and SG stem cell markers

Specific markers for MG stem cells are poorly defined. As MGs share many characteristics with HF-associated SGs we interrogated the snRNA-seq data for known SG stem cell markers and validated the results using RNAscope assays on independent biological samples of adult MG.

Our analysis (Supplementary Fig. S2a, b) showed that *Lrig1* expression in ductular and ductal suprabasal cells was similar to that of the putative MG stem cell marker *Egr2* (Supplementary Fig. S2c, d). We found that acinar basal cells (cluster 16) expressed the SG stem cell markers *Lgr6* (Supplementary Fig. S2a, b; Supplementary Fig. S2f, white arrow) and *Slc1a3*[32] (Supplementary Fig. S2a, b; Supplementary Fig. S1j, white arrows), and also exhibited low-level expression of *Lrig1* (Supplementary Fig. S2a, b; Supplementary Fig. S2e, white arrow) and *Axin2*[33] (Supplementary Fig. S2a, b; Supplementary Fig. S2g, white arrows), along with expression of *Gli2* (Supplementary Fig. S2a, b; Supplementary Fig. S2h, white arrow). Ductular cells (cluster 17) were identified in sectioned tissue as lying at the junction of the acinus and the duct. Ductules were enriched for *Gli2* expression (Supplementary Fig. S2a, b; Supplementary Fig. S2h, yellow arrows) and expressed high levels of *Lrig1* (Supplementary Fig. S2a, b; Supplementary Fig. S2e, yellow arrows), together with low levels of *Lgr6*, *Axin2*, and *Slc1a3* (Supplementary Fig. S2a, b; yellow arrows in Supplementary Fig. S2f, g and Supplementary Fig. S1j). *Slc1a3* expression only localized to the acinus and ductule and was absent from the duct (Supplementary Fig. S2a, b; Supplementary Fig. S1j), like its expression in SGs[32]. The expression of SG stem cell markers in the acinar basal layer and ductule suggested that these markers may also label stem cells within the MG.

## *Lrig1*-, *Lgr6*-, *Axin2*-, *Slc1a3*-, and *Gli2*-expressing cells contribute to MG homeostasis

To determine whether *Lrig1*-, *Lgr6*-, *Axin2*-, *Gli2*- and *Slc1a3*-expressing cells self-renew and contribute to MG homeostasis, we performed lineage tracing assays using *Lrig1-Cre^{ERT2}*, *Lgr6-Cre^{ERT2}*, *Axin2-Cre^{ERT2}*, *Gli2-Cre^{ERT2}* and *Slc1a3-Cre^{ERT2}* mice carrying a *Rosa26^{mTmG}* or *Rosa26^{nTnG}* reporter allele (Fig. 2a). After 2 days of tamoxifen induction, we observed *Lrig1-Cre^{ERT2}*-labeled cells in MG ducts (Fig. 2b, yellow arrowheads) and in acinar basal cells (Fig. 2b, white arrow), in line with expression of endogenous *Lrig1* (Supplementary Fig. S1o). Cells labeled by *Lgr6-Cre^{ERT2}* or *Axin2-Cre^{ERT2}* were present in MG ducts and in the acinar basal layer (Fig. 2c, d), while *Gli2-Cre^{ERT2}*-labeled cells were present in acini (Fig. 2e, white and pink arrowheads), and in the ductules (Fig. 2e, orange arrowhead), but were absent from the duct itself. *Slc1a3-Cre^{ERT2}*-labeled cells were identified in the acinar basal layer (Fig. 2f, white arrowhead) and meibocytes (Fig. 2f, yellow arrowhead) but not in ductules or central ducts. The absence of *Slc1a3-Cre^{ERT2}* labeling in ductules may be due to low *Slc1a3* promoter activity in ductule cells. As the complete maturation process of meibocytes from acinar basal cells in mice takes approximately 9 days[34], we analyzed the descendants of *Lrig1*-, *Lgr6*-, *Axin2*-, *Gli2*- and *Slc1a3*-expressing cells after 90 or 120 days of tracing, providing sufficient time for labeled cells to undergo multiple rounds of cell renewal. We found that progeny of *Lrig1*-, *Lgr6*-, and *Axin2*-expressing cells were present in both MG ducts and acini (Fig. 2b–d), indicating that these cells can self-renew and give rise to differentiating progeny.

Interestingly, at 90 days, GFP+ descendants of *Gli2*-expressing cells not only replenished the ductule (Fig. 2e, orange arrowhead) and

acinus (Fig. 2e, white and pink arrowheads), but were also detected in the central duct (Fig. 2e, yellow and white arrowheads), suggesting that *Gli2*-expressing ductular cells can contribute to the central duct.

*Slc1a3-Cre^{ERT2}*-marked cells were present in the acinar basal layer (Fig. 2f, white arrowhead) and in meibocytes (Fig. 2f, yellow arrowheads), but not in the ductules or central duct. Thus, *Slc1a3-Cre^{ERT2}* marks acinar stem cells but not stem cells that contribute to ductules or central duct.

Taken together, these data demonstrate that *Lrig1*-, *Lgr6*-, *Axin2*-, *Gli2*- and *Slc1a3*-marked populations contain self-renewing stem cells that persist through multiple rounds of MG cell renewal and give rise to differentiated progeny in the duct and/or acinus to maintain MG homeostasis. Furthermore, in line with the prediction from Velocity analysis that the progeny of ductular stem cells can contribute to ducts and acini (Fig. 1c), lineage tracing with *Gli2-Cre^{ERT2}* revealed the presence of descendants of *Gli2*-expressing ductular cells in the central duct.

## *Lrig1*+ stem cells located in the MG ductule migrate toward the acinus

A previous study identified label-retaining cells in ductules, suggesting that ductules may contain stem cells that can replenish the acinus[12]. This hypothesis was supported by Velocity analysis of our snRNA-seq data (Fig. 1c). To test this further, we investigated the behavior of *Lrig1*+ cells in ductules by combining lineage tracing in *Lrig1-Cre^{ERT2} Rosa26^{nTnG}* mice, and ex vivo live imaging. After 40 days of lineage tracing, which is sufficient for the MG to undergo several turnover cycles, we identified GFP+ cell clones in the MG acinus (Supplementary Fig. S3a, b, yellow arrows), ductule (Supplementary Fig. S3a, b, white arrows), and duct (Supplementary Fig. S3a, b, blue arrows). Tarsal plate explants from these mice were cultured ex vivo for 60 minutes, during which period explants remained proliferative and secreted meibum droplets (Supplementary Fig. S4a–c, Supplementary Movie 1). Time-lapse imaging of tarsal plate explants from lineage-traced *Lrig1-Cre^{ERT2} Rosa26^{nTnG}* mice over 15 hours of culture identified GFP+ cells or their daughter cells from division within the ductules that moved toward the acinus and settled in the acinar basal layer (Supplementary Movie 2, arrowheads). Thus, *Lrig1*+ cells in the ductule can contribute to the acinar basal layer. Based on lineage tracing data (Fig. 2), acinar basal stem cells also self-renew, and this is likely the major mechanism for their replenishment.

## Hh signaling maintains MG homeostasis

Given our finding that MG stem cells express similar markers to SG stem cells, we hypothesized that MG and SG stem cell activity are regulated by similar mechanisms. Among several signaling pathways important for SG development and homeostasis[35,36], Hedgehog (Hh) signaling plays a critical role[21]. Interrogation of the snRNA-seq dataset (Supplementary Fig. S5a) and validation using RNAscope (Supplementary Fig. S5b–f) showed that multiple Hh pathway components[37] are expressed in the MG. *Shh* is expressed in a sub-population of acinar basal cells (Supplementary Fig. S5b, green arrows); *Ihh* and *Ptch1* localize to acinar basal cells (Supplementary Fig. S5c, d, green arrows), differentiating meibocytes (Supplementary Fig. S5c, d, white arrow), and ductular cells (Supplementary Fig. S5c, d, yellow arrows); *Smo* is expressed in acinar basal cells (Supplementary Fig. S5e, green arrow), ductular cells (Supplementary Fig. S5e, yellow arrow), and ductal cells (Supplementary Fig. S5e, pink arrow); and *Gli1* is expressed in acinar basal cells (Supplementary Fig. S5f, green arrows), differentiating meibocytes (Supplementary Fig. S5f, white arrow), and ductular cells (Supplementary Fig. S5f, yellow arrow). Additionally, we noted expression of *Ptch1*, *Smo* and *Gli1* in the surrounding stroma (Supplementary Fig. S5d–f, light blue arrows).

To determine whether epithelial Hh signaling is functionally required for adult MG homeostasis, we inducibly deleted the Hh co-

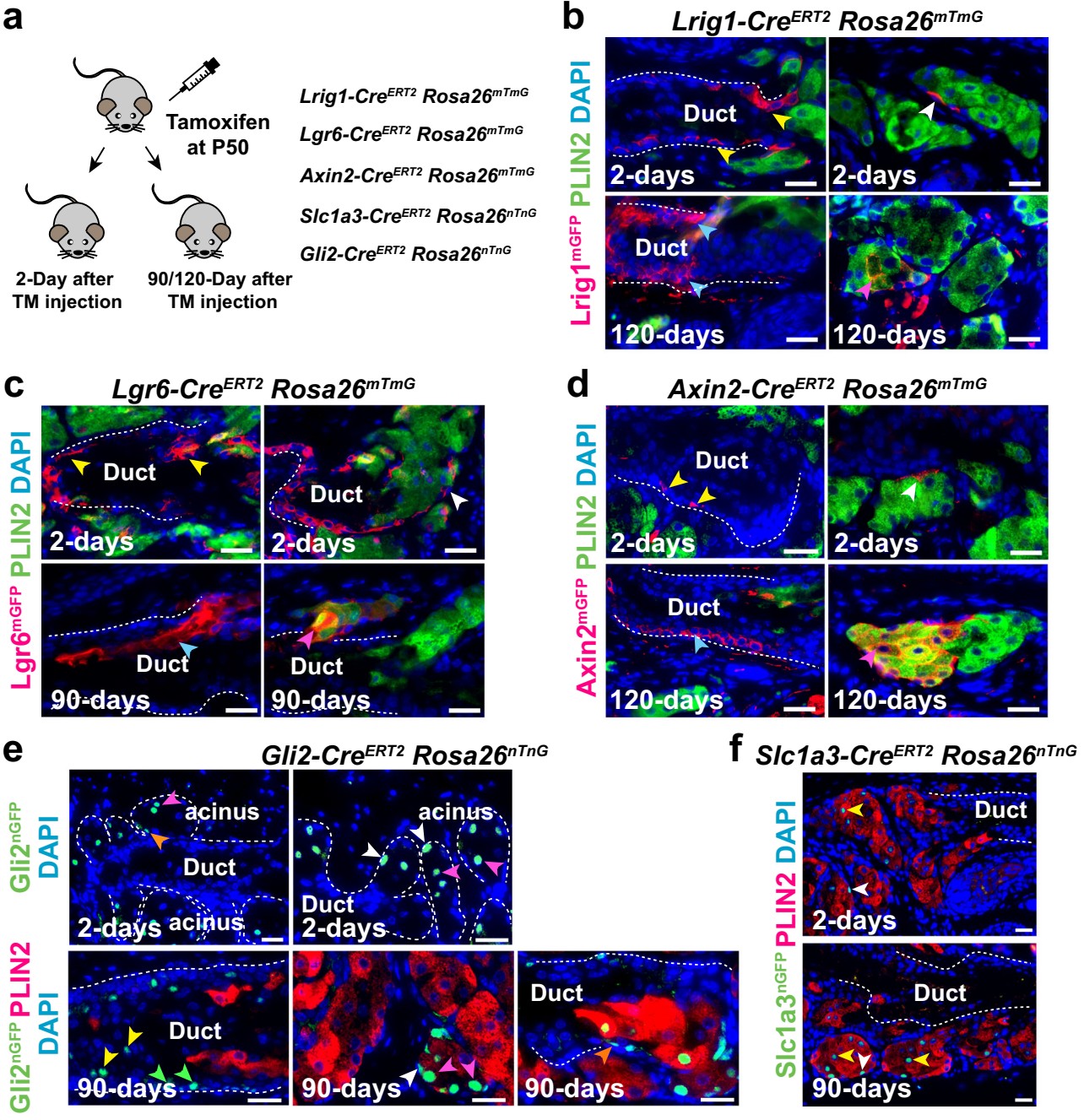

**Fig. 2 | Identification of distinct MG stem cell populations. a** Scheme for lineage tracing. **b** *Lrig1*+ cells in MG ductal basal layer (yellow arrows) and acinar basal layer (white arrow) self-renew and generate progeny that contribute to MG duct (light blue arrows) and acinus (pink arrow). **c** *Lgr6*+ cells in MG duct (yellow arrows) and acinar basal layer (white arrow) self-renew and contribute to MG duct (light blue arrow) and acinus (pink arrow). **d** *Axin2*+ cells in the MG duct (yellow arrows) and acinar basal layer (white arrow) self-renew and replenish MG duct (light blue arrow) and acinus (pink arrow). **e** At 2 days, *Gli2* marks acinar basal cells (white arrowheads), ductule cells (orange arrowhead), and meibocytes (white arrowheads); at 90 days, lineage-traced cells are present in MG duct (yellow and green blue arrowheads), ductule (orange arrowhead) and acinus (white and pink arrowheads). **f** At 2 days, *Slc1a3* marks meibocytes (yellow arrowhead) and acinar basal cells (white arrowhead); at 90 days, lineage tracing labels the acinar basal layer (white arrowhead) and meibocytes (yellow arrowheads). Scale bars: (**b**–**e**), 50 μm; (**f**), 25 μm. White dashed lines in (**b**–**f**) outline MG acini and ducts. *N* = 3 mice (2 males and 1 female) of each genotype were analyzed in lineage tracing experiments for each line and stage. See also Supplementary Figs. S2–S4; Supplementary Movies S1 and S2.

receptor *Smo* in adult MG epithelium by treating *KRT14-Cre*ERT2 *Smo*fl/fl *Rosa26*mTmG mice and *KRT14-Cre*ERT2 *Rosa26*mTmG littermate controls with tamoxifen for 5 days starting at 8 weeks of age. This resulted in reduced size of acini in *Smo*-deleted MGs analyzed at 18- or 29-weeks post-tamoxifen treatment (Fig. 3a–c). Differentiation of MG acini was not obviously affected (Fig. 3d–g), but *Smo*-deleted MGs exhibited significantly decreased proliferation of acinar and ductal basal cells

compared with controls (Fig. 3h–m). Thus, Hh signaling promotes the proliferation of both acinar and ductal basal cells.

## Hyper-activation of Hh signaling in MG epithelium causes expansion of acinar and ductal basal cells

To ask whether Hh signaling is sufficient to promote MG basal cell proliferation, we generated *Krt5-rtTA tetO-GLI2ΔN* (*GLI2ΔN*KrtSrtTA)

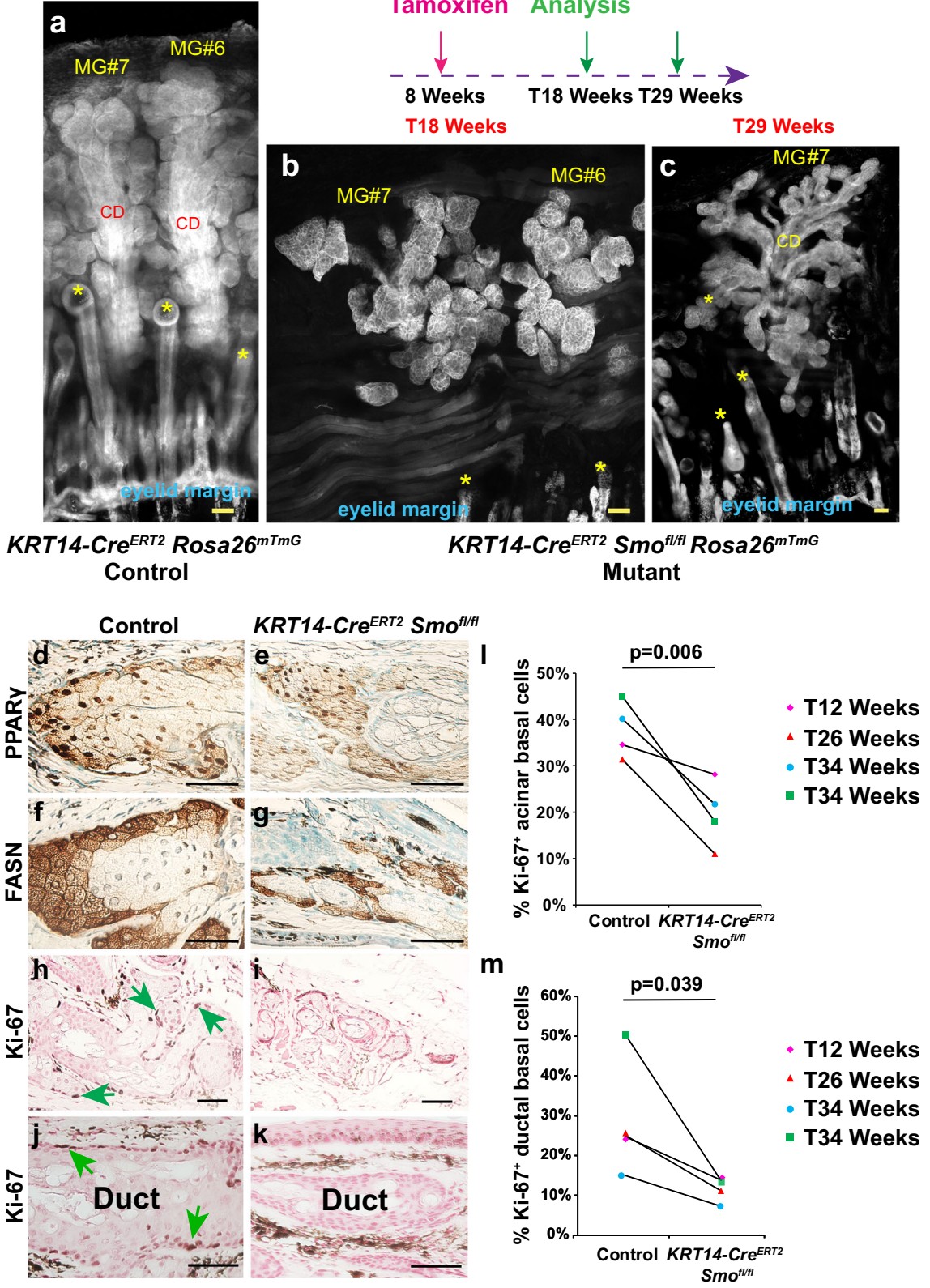

**Fig. 3 | Hh signaling is required for normal levels of MG basal cell proliferation.** **a**–**c** Whole-mount fluorescence images of MG #6 and/or MG #7 in (**a**) *KRT14-Cre^ERT2 Rosa26^mTmG* control mice 18 or 29 weeks after tamoxifen treatment at 8 weeks of age (T18 Weeks and T29 Weeks) (*n* = 4, 2 male and 2 female); **b**, **c** *KRT14-Cre^ERT2 Smo^fl/fl Rosa26^mTmG* mice of T18 Weeks (**b**) or T29 Weeks (**c**) (*n* = 4, 3 male and 1 female). Yellow asterisks indicate hair follicles; CD, MG central duct. (d-g) IHC for PPARγ (**d**, **e**) and FASN (**f**, **g**) showing similar expression levels in control (**d**, **f**) and *Smo*-deficient (**e**, **g**) MGs at 34 weeks after tamoxifen treatment. **h**–**k** IHC for Ki-67 showing reduced acinar and ductal basal cell proliferation in *Smo*-deficient MGs at

34 weeks after tamoxifen treatment (**i**, **k**) compared with control (**h**, **j**, green arrowheads). **l**, **m** Quantitation of the % of Ki-67⁺ cells in the acinar (**l**) and ductal (**m**) basal layer of control and *Smo*-deficient MGs. Littermate pairs were compared. Statistical significance was calculated using a paired two-tailed Student's *t*-test. *n* = 4 *KRT14-Cre^ERT2* control mice (3 male and 1 female) and *n* = 4 *Smo*-deficient (*KRT14-Cre^ERT2 Smo^fl/fl*) mice (3 male and 1 female) were analyzed in (**d**–**m**). At least 70 acinar basal cells and 80 ductal basal cells were analyzed per mouse in (**l**, **m**). Source Data for (**l**, **m**) are provided as a Source Data file. Scale bars: 50 μm. See also Supplementary Fig. S5.

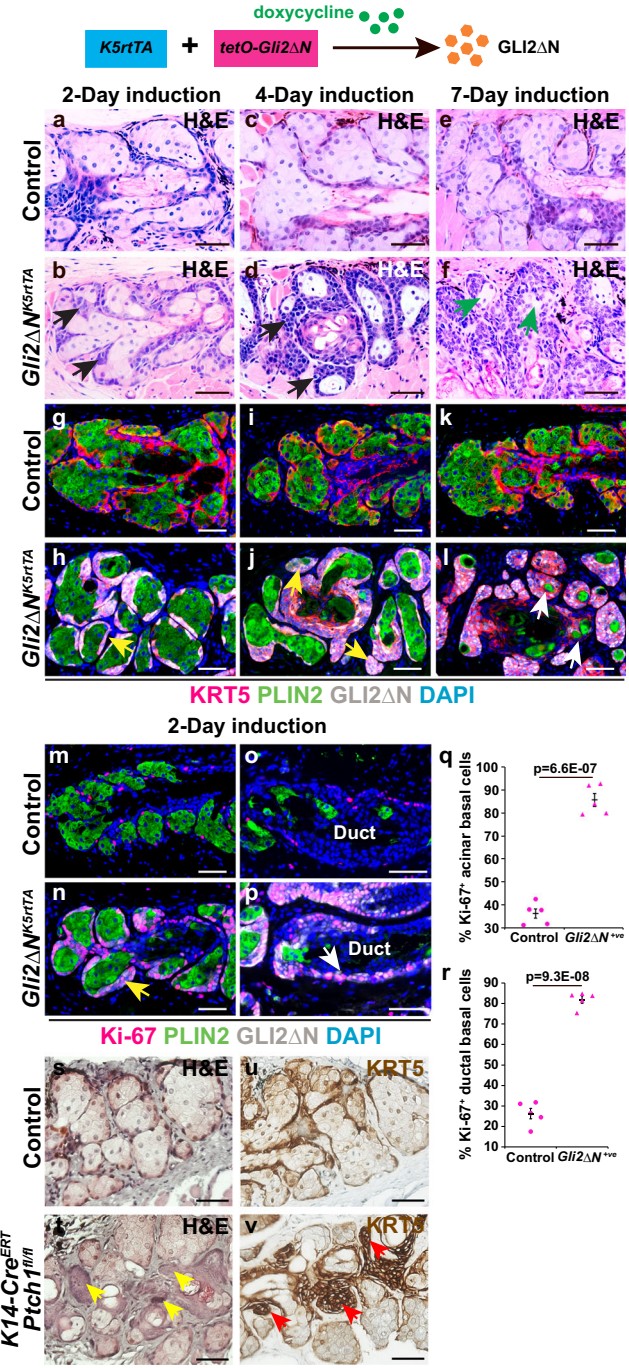

**Fig. 4 | Increased Hh signaling promotes proliferation and expansion of MG basal cells. a–f** Progressive expansion of acinar basal cell clusters (black arrows) in GLI2ΔN-expressing MG after 2 (**a, b**), 4 (**c, d**), and 7 (**e, f**) days of doxycycline treatment at 8 weeks of age. Meibocytes were reduced by 7 days (f, green arrows).
**g–l** Progressive expansion of GLI2ΔN⁺/KRT5⁺ cells (yellow arrows) and reduction of PLIN2⁺ meibocytes (white arrows) in *Gli2ΔN^KrtSrtTA* MG acini after 2 (**g, h**), 4 (**i, j**) and 7 (**k, l**) days of doxycycline treatment. *N* = 3 control mice lacking *Krt5-rtTA* or *tetO-GLI2ΔN* (2 males and 1 female) and *n* = 3 *Gli2ΔN^KrtSrtTA* mice (2 males and 1 female) were analyzed at each time point in (**a–l**); representative samples are shown.
**m–p** Hyperproliferation of GLI2ΔN⁺ cells in *Gli2ΔN^KrtSrtTA* MG acini (m, n, yellow arrow) and ducts (**o, p**, white arrow). **q, r** Quantification of acinar basal cell (**q**) and ductal basal cell (**r**) proliferation. Samples from *n* = 5 control mice (3 males and 2 females) lacking *Krt5-rtTA* or *tetO-GLI2ΔN* and samples from n = 5 *Gli2ΔN^KrtSrtTA* mice (3 males and 2 females) were analyzed in (**m–r**). Statistical significance in (**q, r**) was calculated with unpaired two-tailed Student's t-test. Data are presented as mean +/− SEM. At least 100 acinar cells and 110 ductal cells were analyzed from each animal. Source Data for (q, r) are provided as a Source Data file. **s, t** Overgrowth of *Ptch1*-deficient acinar basal cells (yellow arrows) in *KRT14-Cre^ERT2 Ptch1^fl/fl* mice 12 weeks after tamoxifen treatment at 8 weeks of age. **u, v** Expansion of KRT5⁺ cells in *Ptch1*-deficient acini (red arrows). Independent biological samples from *n* = 3 male *KRT14-Cre^ERT2 Ptch1^fl/fl* mice, *n* = 2 age-matched male controls and *n* = 2 age-matched female controls were analyzed in (**s–v**). Representative images are shown. Scale bars: 50 µm.

As GLI2ΔN is not subject to the same regulation as endogenous GLI2, it might produce off-target effects that differ from those resulting from loss of endogenous Hh signaling. To control for this, we examined the effects of inducible epithelial deletion of the endogenous *Ptch1*, Hh inhibitory receptor, in the MGs of adult *KRT14-Cre^ER Ptch1^fl/fl* mice 12 weeks after induction. Consistent with the effects of forced expression of GLI2ΔN, we found that loss of epithelial *Ptch1* resulted in overgrowth of KRT5⁺ acinar basal cells, with a lesser effect on ductal compared with acinar morphology (Fig. 4s-v). Thus, hyper-activation of Hh signaling, either via forced expression of GLI2ΔN, or through deletion of endogenous epithelial *Ptch1*, caused basal cell expansion.

### GLI2ΔN promotes proliferation of *Lrig1*- and *Lgr6*-expressing cells at the expense of differentiation

To delineate the mechanisms responsible for GLI2ΔN-mediated acinar basal cell expansion, we performed bulk RNA-seq on MGs laser-captured from control and *GLI2ΔN^KrtSrtTA* mice after 4 days of doxycycline treatment (Fig. 5a). Gene ontology (GO) enrichment analysis showed that the most significantly upregulated pathways were those involved in regulation of cell proliferation, including regulators such as *Ki67* and *Ccnd1* (Fig. 5b, c), in line with IF results (Fig. 5d–g). Expression of Hh target genes *Gli1*, *Ptch1*, and *Hhip* was also significantly elevated in GLI2ΔN-expressing MGs compared to controls (Fig. 5c). The most significantly downregulated genes were those associated with control of lipid metabolism (Fig. 5b) such as *Pparg*, and meibocyte differentiation including *Scd3*, *Scd4*, and *Plin2* (Fig. 5c). RNAscope and IF analyses confirmed increased expression of *Gli1* and *Ccnd1* and reduced levels of PPARγ and PLIN2 expression in GLI2ΔN-expressing MGs compared to controls (Fig. 5d-i). Notably, cell populations expressing MG stem cell markers *Lrig1* and *Lgr6* were expanded upon GLI2ΔN expression in MGs (Fig. 5c, j–m). Collectively, these data suggest that forced GLI2ΔN expression in MG epithelium promotes proliferation, expands the number of cells expressing *Lrig1* and *Lgr6*, and, either directly or indirectly, suppresses meibocyte differentiation.

### *Lrig1*-, *Lgr6*- and *Axin2*-expressing MG stem cells contribute to MG overgrowth driven by GLI2ΔN

To test whether *Lrig1*-, *Lgr6*- and *Axin2*-expressing MG stem cells contribute to basal cell expansion caused by forced GLI2ΔN expression, we performed lineage tracing in *Lrig1-Cre^ERT2 Rosa26^mTmG*, *Lgr6-Cre^ERT2 Rosa26^mTmG*, and *Axin2-Cre^ERT2 Rosa26^mTmG* mice carrying *Gli2ΔN^KrtSrtTA*. Adult mice were treated with tamoxifen to induce Cre activity; after 30 days, mice were placed on doxycycline water to

transgenic mice that expressed an activated form of GLI2 (GLI2ΔN) in basal epithelial cells, including acinar and ductal basal cells in the MG, in a doxycycline-dependent manner[38,39]. GLI2ΔN⁺ acinar basal cells emerged within 2 days of initiating doxycycline treatment (Fig. 4a, b, g, h), and began to expand, starting to replace differentiating PLIN2⁺ meibocytes by 4 days after induction (Fig. 4c, d, i, j). After 7 days, most meibocytes had been replaced by GLI2ΔN⁺ cells (Fig. 4e, f, k, i) that showed statistically significantly increased proliferation frequency compared to control acinar basal cells (Fig. 4m, n, q). Similarly, forced expression of GLI2ΔN⁺ caused a significant increase in basal cell proliferation frequency in MG ducts, although the effects of GLI2ΔN on duct morphology were less pronounced than its effects in the acini (Fig. 4o, p, r).

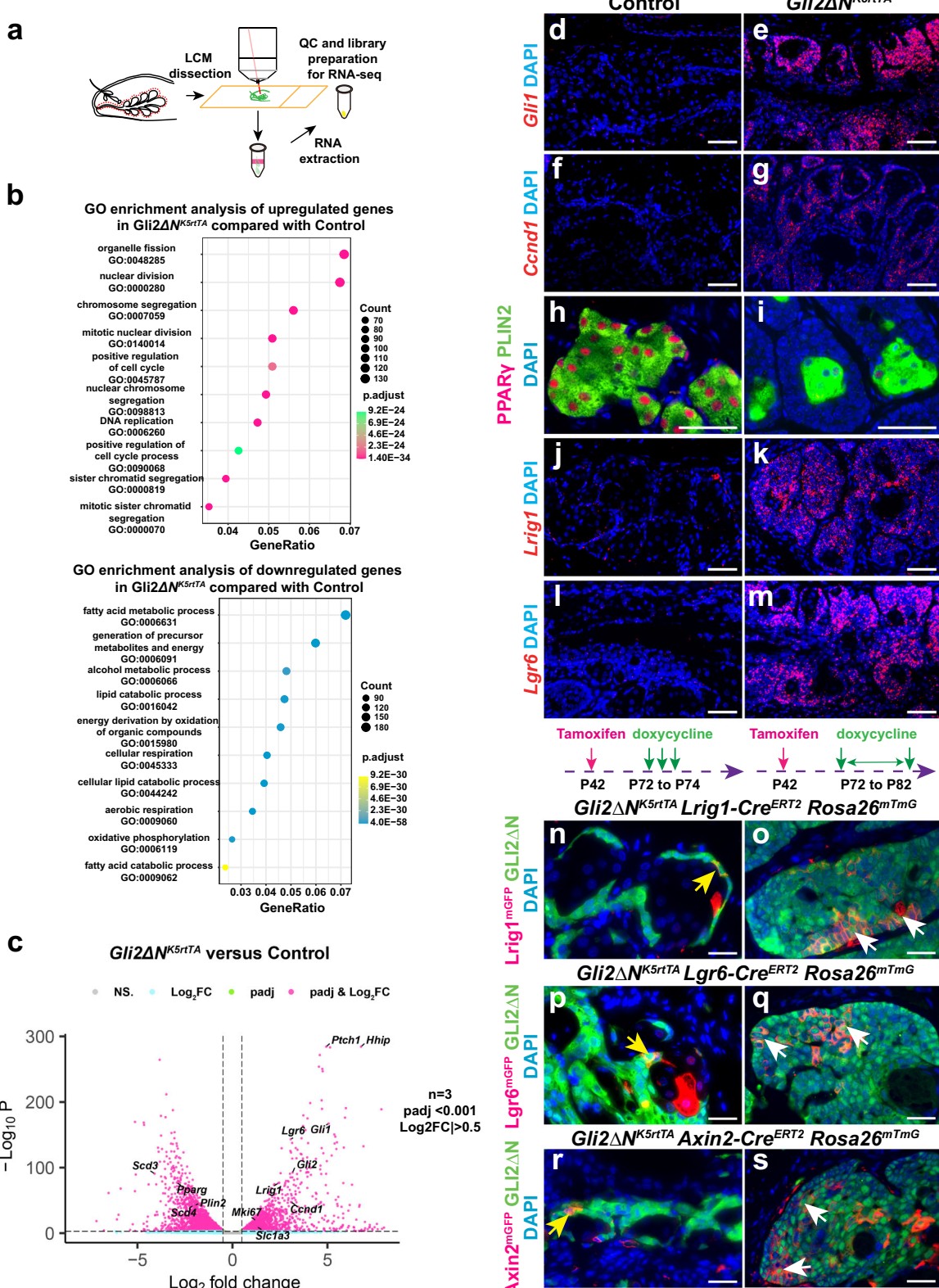

induce GLI2ΔN expression in MG stem cells. Two days after initiating doxycycline treatment, GFP+/GLI2ΔN+ cells were present in the acinar basal layer in each of these lines (Fig. 5n, p, r, yellow arrows). After 10 days of doxycycline treatment, we observed expansion of GFP+/GLI2ΔN+ clones in all three lines (Fig. 5o, q, s, white arrows). These observations indicate that GLI2ΔN-expressing MG stem cells contribute to MG basal expansion.

## GLI2 is broadly expressed and *LRIG1* and *LGR6*-expressing undifferentiated cells are expanded in human Meibomian gland carcinoma

Abnormal stem cell activity is frequently observed in cancers[40,41], but the role of stem cells in human Meibomian gland carcinoma (MGC) is unknown. To determine whether expanded GLI2 and stem cell marker expression are features of human MGC, we analyzed 7 control MG

**Fig. 5 | Forced GLI2ΔN expression expands MG stem cell populations and impedes meibocyte differentiation. a** Scheme for bulk RNA-seq of laser-captured MG samples. **b** GO enrichment analysis of bulk RNA-seq data from *GLI2ΔN^KrtSrtTA^* and littermate control MGs 4 days after induction at 8 weeks of age showing the top 10 enriched pathways. **c** Volcano plot showing genes upregulated or downregulated at 4 days. Independent biological samples from *n* = 3 *GLI2ΔN^KrtSrtTA^* mice (2 male and 1 female) and independent biological samples from n = 3 littermate control mice lacking *Krt5-rtTA* or *tetO-GLI2ΔN* (2 males and 1 female) were analyzed; differentially expressed genes were defined as padj<0.001 and Log2FC < -0.5 or Log2FC > 0.5. FDR calculation was performed by DESeq2 v1.20.0 with the Benjamini-Hochberg procedure. **d–g** RNAscope showing upregulation of *Gli1* and *Ccnd1* in *GLI2ΔN^KrtSrtTA^* MGs. **h, i** IF data showing decreased PPARγ and PLIN2 expression in *GLI2ΔN^KrtSrtTA^* acini. (**j-m**)

RNAscope showing expanded *Lrig1* and *Lgr6* expression in *GLI2ΔN^KrtSrtTA^* MGs. Independent samples from 3 *Gli2ΔN^KrtSrtTA^* mice (2 males and 1 female) and 3 littermate controls of genotypes *tetO-GLI2ΔN* or *Krt5-rtTA* (2 males and 1 female) were used for RNAscope and IF; all mice were doxycycline-treated for 4 days. (n-s) *GLI2ΔN^KrtSrtTA^ Rosa26^mTmG^* mice carrying inducible Cre alleles driven by *Lrig1* (**n, o**), *Lgr6* (**p, q**) or *Axin2* (**r, s**) promoters were tamoxifen-induced at P42 to induce Cre activity and placed on oral doxycycline at P72 to induce GLI2ΔN expression. mGFP expression (red signal) and GLI2 expression (green signal) were analyzed by IF at P74 (**n, p, r**) or P82 (**o, q, s**). GLI2ΔN-expressing cells in the acinar basal layer positive for *Lrig1* (**n**), *Lgr6* (**p**), or *Axin2* (**r**) (yellow arrows) give rise to clones that contribute to MG overgrowth (**o, q, s**, white arrows). *n* = 3 (1 male and 2 females) samples were analyzed per line per time point. Scale bars: (**d–m**), 50 μm; (**n–s**), 25 μm.

---

samples and 10 MGC samples. The MGC samples showed loss of an organized structure, and MG ducts and ductules were not apparent in the tumor area (Fig. 6a–l). Central cells in the MGC samples exhibited downregulation of KRT14 (Fig. 6b, j, l) compared with control MGs (Fig. 6a, i, k); loss of KRT14 expression may reflect epithelial-mesenchymal transition of the tumor cells.

We found that in all control MGs examined, GLI2 was predominantly expressed in acinar basal cells (Fig. 6a, yellow arrows) and differentiating meibocytes (Fig. 6a, white arrows) but was absent in fully differentiated meibocytes (Fig. 6a, light blue arrows). By contrast, GLI2 was broadly expressed in all 10 MGC samples (Fig. 6b). *GLI1* was weakly expressed in acinar basal cells of control MGs (Fig. 6c, yellow arrows), but was readily detected in 7 of 10 MGC samples (Fig. 6d). *LRIG1* was predominantly expressed in the acinar basal layer of control MGs (Fig. 6e, yellow arrows), but was broadly and strongly expressed in all 10 MGC samples (Fig. 6f). Similarly, *LGR6* expression was detected in acinar basal cells of control MGs (Fig. 6g, yellow arrow), but was broadly expressed in MGC samples (9/10) (Fig. 6h). In control MGs, proliferation was restricted to the basal layer of the acinus (Fig. 6i, white arrows), which lacked expression of differentiation marker PLIN2 (Fig. 6k, white arrows). However, in MGC samples, proliferative cells were widely present (Fig. 6j, yellow arrows) and PLIN2 expression was undetectable in most tumor cells (Fig. 6l, yellow arrows). These data indicate that human MGC is characterized by the expansion of poorly differentiated and proliferative tumor cells that express MG stem cell markers.

### Aged MGs display fewer cells expressing Hh pathway genes

The mechanisms underlying MG dropout in aging remain poorly understood. As previously reported[9,42], we found that in aged (21-month) versus young (8-week) MGs, intracellular distribution of PPARγ was altered, being relatively reduced in the cytoplasm and enriched in the nucleus in aged samples (Supplementary Fig. S6a, b), while the percentage of KRT5+ acinar basal cells that were positive for the proliferation marker Ki-67 was significantly decreased in aged compared with young MGs (Supplementary Fig. S6c–e).

As our data revealed a key role for Hh signaling in acinar basal cell proliferation, we asked whether expression of Hh pathway genes was altered in aging. Analysis of snRNA-seq data from young and aged MGs using CellChat, which assesses intercellular communication based on patterns of ligand and cognate receptor expression[43], predicted that in young mice, IHH ligands secreted from MG ductal cells and differentiating Meibocytes are received by ductular cells, acinar basal cells, and dermal cells (Supplementary Fig. S6f, g). These analyses suggest that Hh ligands act in a paracrine fashion in young MGs, but do not exclude autocrine mechanisms. In aged mice, the levels of Hh pathway gene expression were decreased to the point that significant Hh-mediated interactions could not be identified (Supplementary Fig. S6h, i). Violin plots derived from snRNA-seq data showed that the percentages of ductular cells expressing *Gli2* and acinar basal cells expressing *Ptch1* mRNA were reduced in aged versus young MGs (Fig. 7a). Validation by RNAscope revealed that there were fewer cells

expressing *Ptch1* mRNA in the acinar basal layer, differentiating meibocyte population, and stroma in aged MG (Fig. 7c) compared with MG at 8 weeks (Fig. 7b). Additionally, there were fewer cells expressing *Gli2* mRNA in aged compared with young MG, especially in the ductules (Fig. 7d, e, white arrows). By contrast, expression of the stem cell markers *Lrig1*, *Lgr6*, *Slc1a3*, and *Axin2* in ductular or acinar basal cells did not differ appreciably in aged compared with young MGs (Supplementary Fig. S6j, k). These data may suggest that a specific *Gli2*+ stem cell population in ductules is reduced in aging; alternatively, as Hh signaling can upregulate expression of *Gli2* as well as *Ptch1* mRNA[44], the reduced number of *Gli2*+ cells may reflect decreased signaling through the Hh pathway.

### Association of GLI2 with acetylated lysine is increased in aged MGs

Despite reduced Hh pathway activity, expression of Hh ligands was maintained in aged MG (Fig. 7f, g), suggesting that mechanisms other than decreased ligand expression downregulate Hh signaling in aged MGs. While fewer cells expressing *Gli2* were present in aged compared with young MGs (Fig. 7d, e), proximity ligation assays (PLA) revealed that subsets of the cells that still expressed GLI2 protein in aged MG and surrounding stroma displayed its elevated association with acetylated lysine, both within the acini (Fig. 7h, i, white arrows) and in the surrounding stroma (Fig. 7h, i, blue arrows). Acetylation mitigates the transcriptional activity of GLI2[45]; thus, this mechanism could potentially account for the observed decrease in the percentages of cells expressing Hh-regulated genes, including *Ptch1* and *Gli2* itself, in MGs and their stromal environment.

MGs with epithelial deletion of the histone deacetylases HDAC1 and HDAC2 exhibit reduced proliferation of acinar basal cells[46]. As GLI2 can be deacetylated by HDAC1[46], we hypothesized that association of GLI2 with acetylated lysine would be increased in the absence of HDAC1/2. To test this, we induced *Hdac1/2* deletion in KRT5+ MG basal cells of *Krt5-rtTA tetO-Cre Hdac1^fl/fl^ Hdac2^fl/fl^* mice; as KRT5+ acinar and ductal basal stem cells give rise to differentiating acinar and duct cells, this approach results in *Hdac1/2* deletion in the entire epithelial MG within 9 days of induction[46]. *Hdac1/2*-deleted acinar basal and differentiating cells in MGs from 8-week-old mice that had been doxycycline-treated for 10 days, showed increased PLA signals for GLI2 and acetylated lysine (Fig. 7j, k, white and yellow arrows). PLA signals in the surrounding stromal cells, where *Hdac1/2* were not deleted, were similar in mutant and control mice, providing an internal control for the PLA assay (Fig. 7j, k, blue arrows). Furthermore, *Ptch1* expression was attenuated within HDAC1/2-deficient acini (Fig. 7l, m, white and yellow arrows) but not in the surrounding stroma (Fig. 7l, m, blue arrows). Violin plots derived from snRNA-seq data revealed similar numbers of ductular cells expressing *Hdac1* and increased numbers expressing *Hdac2* in aged versus young MGs; and decreased numbers of acinar basal cells expressing *Hdac1* and increased numbers expressing *Hdac2* in aged versus young MGs (Fig. 7n). As deletion of *Hdac1* or *Hdac2* alone does not impact MG morphology[46], these changes are unlikely to account for age-related MG phenotypes.

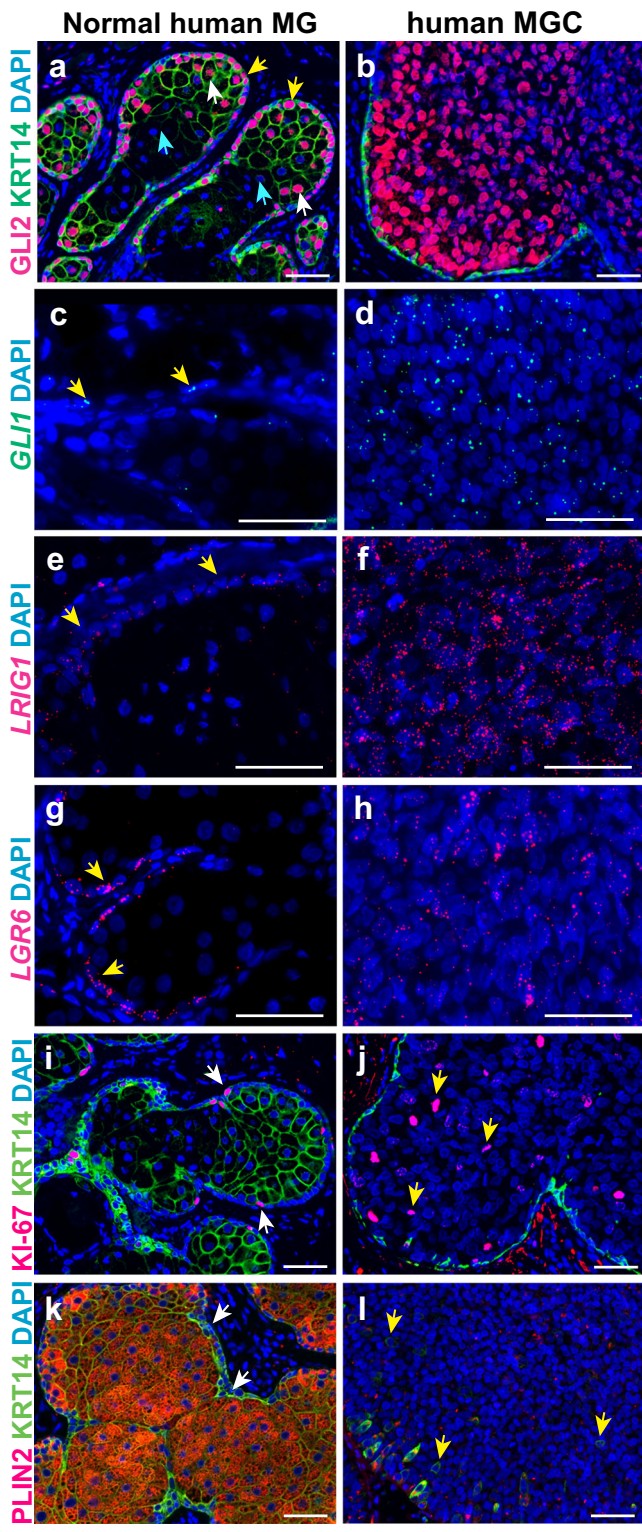

**Fig. 6 | GLI2-, *LRIG1*- and *LGR6*-expressing undifferentiated cells are expanded in human MGC. a, b** In normal human MG GLI2 protein localizes to acinar basal cells (**a**, yellow arrows) and differentiating meibocytes (**a**, white arrows) but is absent from fully differentiated meibocytes (**a**, light blue arrows); in MGC samples GLI2 is broadly expressed (**b**). **c, d** In normal human MG, *GLI1* mRNA is weakly expressed in some acinar basal cells (**c**, yellow arrows); *GLI1*⁺ cells are widely present in MGCs (**d**). **e–h** *LRIG1*⁺ and *LGR6*⁺ cells are primarily present in the acinar basal layer of normal human MGs (**e, g**, yellow arrows); MGC displays expansion of *LRIG1*⁺ and *LGR6*⁺ cells (**f, h**). **i, j** Ki-67⁺ cells localize to the normal human acinar basal layer (**i**, white arrows) and are expanded in MGC (**j**, yellow arrows). **k, l** PLIN2-/KRT14⁺ cells are restricted to the acinar basal layer in normal human MG (**k**, white arrows) but are distributed more broadly in MGC tissue (**l**, yellow arrows). 7 normal human MG and 10 human MGC samples were analyzed; representative data are shown. Scale bars: 50 µm.

## Decreased HBEGF signaling is associated with MG aging

Abrogation of epithelial Hh signaling in adult life did not result in complete loss of MGs, suggesting important contributions of additional signaling pathways to MG maintenance and the loss of these functions in aging. CellChat analysis of snRNA-seq data from young and aged MGs predicted that the EGF pathway contributes to intercellular communication among MG subpopulations and between MG epithelial cells and the surrounding dermal niche and is attenuated in aged mice (Fig. 8a, b). CellChat analysis specifically predicted that HBEGF-EGFR-mediated crosstalk between dermal cells and ductular and acinar basal cells is abolished in aged MG (Fig. 8c, green pathways). Analysis of snRNA-seq data showed that the relative percentages of cells expressing *Hbegf* were reduced in aged versus young differentiating meibocyte, acinar basal, and dermal fibroblast populations (Fig. 8d). These data were validated via RNAscope analysis of independent samples that revealed fewer *Hbegf*-expressing cells and decreased *Hbegf* expression levels per cell in the aged versus young acinar basal layer (Fig. 8e, yellow arrows) and stroma (Fig. 8e, white arrows). In line with this, phosphorylation of ERK1/2 (p-ERK1/2), which can be stimulated by EGFR activation[47], was decreased in aged MG (Fig. 8f). Taken together with known pro-proliferative functions for EGF signaling in cultured human or rabbit MG epithelial cells in vitro[48,49], and in mouse eyelid and MG morphogenesis in vivo[50], these results suggest that the HBEGF-EGFR axis is important for MG homeostasis and that its decreased activity contributes to reduced proliferation of acinar basal cells in aged MG. Other pathways predicted to be dysregulated in aging included FGF signaling (Fig. 8a, b), consistent with the role of FGF signaling in sebaceous gland regeneration[16], and results from bulk RNA-seq of young and old tarsal plates[51].

## Aging disrupts the MG microenvironment

To delineate additional mechanisms that may contribute to the aging of MG, we performed GO analysis of differentially expressed genes in snRNA-seq data from young and aged MGs. Unexpectedly, the results showed statistically significant aging-related decreases in the expression levels of neural guidance genes such as *Slit3*, *Robo2*, *Sema3a* and *Epha7* in the MG acinar basal layer and surrounding dermal cells (Fig. 9a, b, e, f). These data were validated by RNAscope (Fig. 9c, d). These findings, together with the known impact of sensory nerves on MG epithelial cells[52], and decreased peripheral innervation of the lacrimal gland in aged mice[53,54], suggested that altered peripheral nerve function could contribute to MG defects in aging. Consistent with this, IF for the pan-neural marker PGP9.5[28] revealed that the density of nerves near MG acinar basal cells was compromised in aged compared to young mice (Fig. 9g–i).

In addition to the neuron-regulatory genes, downregulated DEGs in aged eyelid dermal fibroblasts included those responsible for collagen synthesis and extracellular matrix assembly, such as *Col1a1*, *Col1a2* and *Col3a1* (Fig. 9j–m), in line with previous data indicating disrupted collagen synthesis in aged dermal fibroblasts[55].

Furthermore, IF assays failed to detect an appreciable difference in the expression of HDAC1 or HDAC2 proteins between young and aged MG acini (Fig. 7o-r). Thus, mechanisms that affect HDAC1/2 activity, rather than altered expression levels of these proteins, may account for increased association of GLI2 with acetylated lysine in aged MGs.

Taken together these observations suggest that the decreased percentages of cells exhibiting Hh activity in aged MG and the surrounding stroma may result in part from increased levels of GLI2 acetylation that mitigate its transcriptional activity in subsets of acinar and stromal cells.

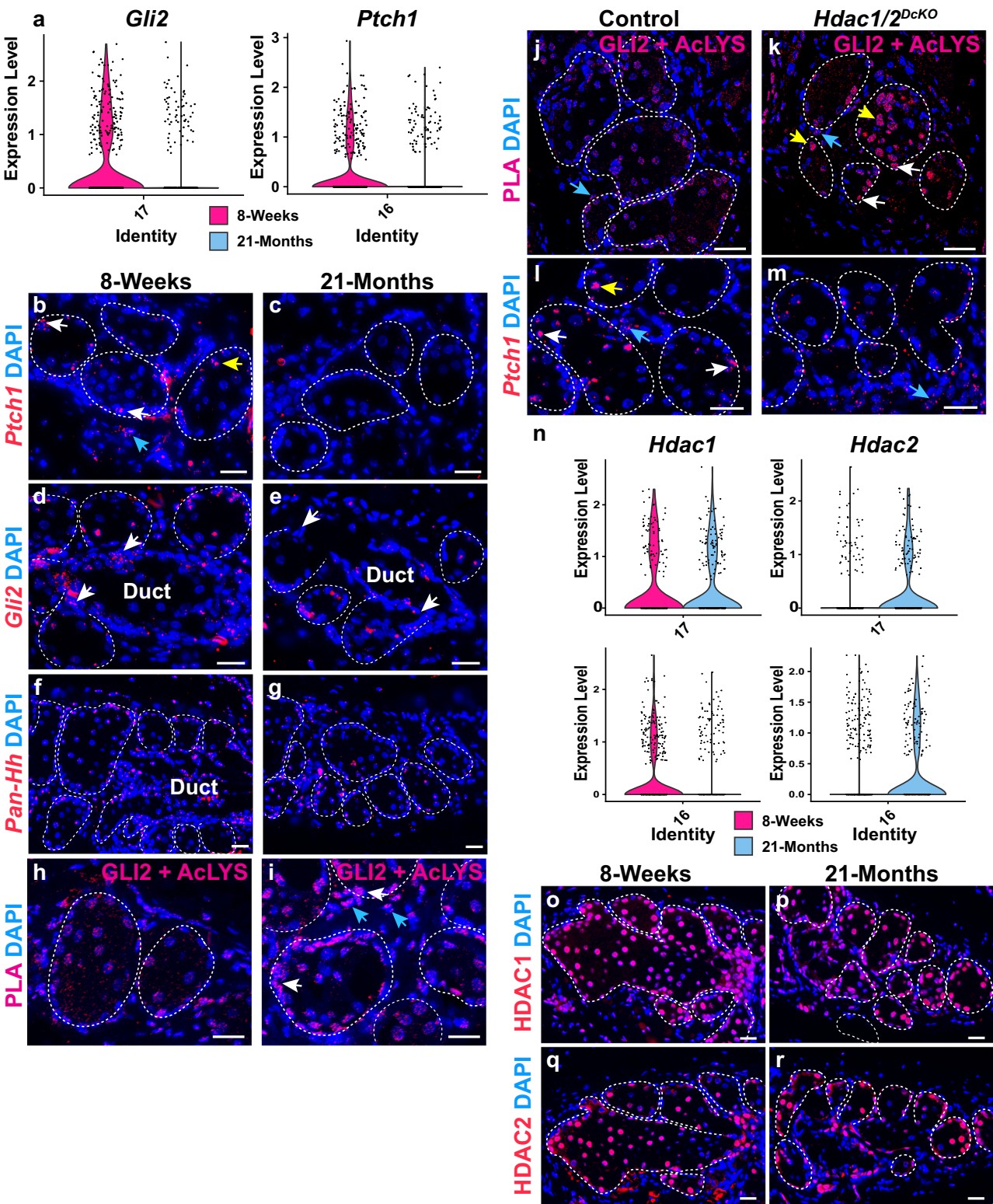

By contrast, upregulated DEGs in aged MG acinar basal cells and ductular cells were associated with oxidative and other forms of stress (Supplementary Fig. S7a–d), which contribute to aging in other contexts[56,57], while upregulated DEGs in aged MG dermal cells were associated with inflammation, consistent with immune cell infiltration in aged MGs[6] (Supplementary Fig. S7e–h).

Taken together, our analyses of aged versus young MGs revealed that MG aging is associated with multiple potential mechanisms, including decreased Hh and EGF signaling, altered function of peripheral neurons, a disrupted dermal microenvironment, and inflammation (Fig. 10).

## Discussion

Little is known about the molecular identities, locations, and behavior of MG stem cells, the signaling pathways that control their activity in homeostasis and aging, and the nature of their tissue niche. Significant findings of the current study include the identification of markers for specific populations of stem cells that maintain distinct regions of the

**Fig. 7 | Aged MGs exhibit fewer cells with Hh activity and elevated association of GLI2 with acetyl-lysine. a** Violin plots derived from snRNA-seq data show relatively reduced percentages of cells expressing *Gli2* and *Ptch1* in the ductule and acinar basal layer, respectively, in aged compared to young MGs. **b, c** *Ptch1* is expressed in acinar basal cells (**b**, white arrow), some meibocytes (**b**, yellow arrow), and surrounding stromal cells (**b**, blue arrow) at 8 weeks; fewer *Ptch1*-expressing cells are present in aged MG (**c**). (**d, e**) Fewer *Gli2*-expressing cells are present in aged MG particularly in the ductule (**d, e**, white arrows). **f, g** RNAscope using a pan-*Hh* probe which detects both *Ihh* and *Shh* mRNAs reveals similar expression of mRNA for Hh ligands in the acini of young (**f**) and aged (**g**) MGs. **h, i** PLA for GLI2 and acetyl-lysine shows that close association (<40 nm) of GLI2 with acetyl-lysine is elevated in subsets of cells within aged (**i**) compared with young (**h**) MG acini (**i**, white arrows) and stroma (**i**, blue arrows). 3 male C57BL/6 J mice were analyzed at each age in (**b-i**); representative data are shown. **j, k** Association of GLI2 and acetyl-lysine is elevated in the acinar basal layer (**k**, white arrows) and meibocytes

(**k**, yellow arrows) but not stromal cells (**j**, **k**, blue arrows) in *Krt5-rtTA tetO-Cre Hdac1^fl/fl Hdac2^fl/fl* (*Hdac1/2^DcKO*) mice doxycycline induced from P48 and analyzed at P58 compared to control mice of genotype *Hdac1^fl/fl Hdac2^fl/fl* and lacking *Krt5-rtTA* or *tetO-Cre*. (**j**). **l, m** *Ptch1* expression in control acinar basal cells (**l**, white arrows) and differentiating acinar cells (**l**, yellow arrow) is reduced in HDAC1/2-deficient MG acini (**m**); stromal *Ptch1* expression is similar in mutants and controls (**l, m**, blue arrows). 3 male mice of each genotype were analyzed in (**j-m**); representative data are shown. **n** Violin plots of snRNA-seq data indicate that there is a relatively lower percentage of *Hdac1*-expressing acinar basal cells (cluster #16) but a higher percentage of *Hdac2*-expressing acinar basal cells and ductular cells (cluster #17) in aged compared to young MG. **o–r** IF for HDAC1 (**o, p**) and HDAC2 (**q, r**) shows similar expression in young (**o, q**) and aged (**p, r**) MGs. 3 male C57BL/6 J mice of each age were analyzed in (**o–r**); representative data are shown. White dashed lines in (**b–m**; **o–r**) outline MG acini and/or ducts. Scale bars: 25 μm. See also Supplementary Fig. S6.

MG; evidence that ductules harbor stem cells for both duct and acinar structures; a key role for Hh signaling in MG stem cell proliferation; decreased Hh and EGF signaling in aged compared with young MGs, suggesting possible therapeutic approaches to enhance MG proliferation in aging; and critical alterations in the MG stromal niche during aging, including in collagen expression and nerve density.

Prior data showed that distinct basal cell populations replenish the central MG ducts and the acini, but specific markers for these stem cell sub-populations had not been identified. We found that basal stem cells marked by *Slc1a3-Cre^ERT2* exclusively contribute to the acinus; by contrast, lineage tracing analyses revealed that both duct and acinus were replenished by stem cells marked by *Lrig1-Cre^ERT2*, *Lgr6-Cre^ERT2*, *Axin2-Cre^ERT2* or *Gli2-Cre^ERT2*. Further studies will be needed to identify specific markers for stem cell populations that exclusively renew MG ductules and ducts.

MG ductules, which connect the central duct and the acini, contain label-retaining cells and have been suggested to contribute to MG homeostasis[12]. However, experimental evidence for this concept has been lacking. Our snRNA-seq analysis revealed that ductules express genes that are characteristic of both the MG duct (*Krt17* and *Egr2*) and the acinus (*Pparg* and *Slc1a3*). Velocity analysis of the snRNA-seq data predicted that ductular cells contribute to both the central duct and the acinus. In line with this, we found that ductular cells express *Slc1a3* as well as *Lrig1*, *Lgr6*, *Axin2*, and *Gli2*. While *Slc1a3-Cre^ERT2* did not label ductules, likely due to low levels of *Slc1a3* promoter activity in ductule cells, lineage tracing combined with live imaging of MG explants showed that LRIG1⁺ cells in the ductule can migrate towards the acinus. Furthermore, *Gli2* expression is absent in the central duct, but ductular cells lineage-traced for *Gli2* populate the central duct. Taken together, these results indicate that the ductule harbors stem cells for both duct and acinus.

Our snRNA-seq data and analyses of mice lacking the Hh receptor Smo revealed that Hh signaling plays a key role in promoting the proliferation of MG stem cells. Consistent with this, forced expression of activated GLI2 or inducible deletion of *Ptch1* in MG epithelial cells resulted in MG basal cell expansion. We found that like in mouse GLI2ΔN-expressing MGs, human GLI2⁺ MGC cells are proliferative and poorly differentiated, exhibiting broad and increased expression of the stem cell markers *LRIG1* and *LGR6*, suggesting that they arise from uncontrolled stem cell expansion. To date, genetic mutations of Hh components have not been identified in MGC, suggesting that increased Hh signaling, while contributing to MGC proliferation, may not be a primary driver of MGC. In line with this, forced expression of GLI2ΔN or deletion of epithelial *Ptch1* resulted in stem cell proliferation and basal cell expansion in our mouse models, but was not sufficient to produce histological indications of MGC[58] in the time frames of our studies. Further investigation is required to delineate the

mechanisms underlying MGC and the contributions of Hh signaling to this disease.

Interestingly, although MG basal cell proliferation was reduced in aging, expression of the stem cell markers *Lrig1*, *Lgr6*, *Slc1a3*, and *Axin2* was not appreciably altered in aged versus young MG ductular and acinar basal cells. By contrast, aged versus young MGs had fewer cells expressing *Gli2*, which we found also marks MG stem cells. As *Gli2* can be upregulated by Hh signaling[44], the decline in *Gli2* mRNA expression may reflect the decrease in Hh pathway activity that we observed in aged MGs. Alternatively, it is possible that a specific *Gli2*⁺ stem cell population is reduced in aging. Further work will be needed to distinguish these mechanisms.

The decline in Hh pathway activity in aged MG could contribute to aging-associated reduced proliferation and MG dropout. Expression levels of Hh ligands were similar in young and aged MGs; however, while there were fewer cells expressing *Gli2* mRNA in aged MGs, those cells with remaining GLI2 protein displayed increased association of GLI2 with acetylated lysine, which dampens its transcriptional activity and could result in decreased expression of Hh-regulated genes including *Gli2* itself. GLI2 is deacetylated by the histone deacetylases HDAC1/2, and we found that the association of GLI2 with acetylated lysine was increased in HDAC1/2-deficient MGs. Consistent with this, we previously showed that depletion of epithelial HDAC1/2 causes decreased MG basal cell proliferation[46]. Taken together, these data support a model in which decreased Hh signaling in aging results in part from altered HDAC1/2 activity that impacts GLI2.

Deletion of epithelial *Smo* did not result in complete loss of MGs, indicating the existence of additional critical pathways. Analysis of snRNA-seq data using CellChat predicted the existence of HBEGF signaling between dermal fibroblasts and both acinar and ductular basal cells that was strongly decreased in aging. This prediction was validated by analysis of *Hbegf* and p-ERK1/2 expression in independent samples of young and aged MGs and is consistent with in vitro experiments showing that EGFR activation is sufficient to stimulate the proliferation of MG epithelial cells[48,49]. EGFR signaling displays extensive crosstalk with the Hh pathway in other cellular contexts; for instance, Shh modulates EGFR-dependent proliferation in embryonic stem cells, neural stem cells, and keratinocytes by transactivating EGFR[59–61]. Further research will be needed to determine the precise role of HBEGF-EGFR signaling, its interactions with Hh signaling, and the mechanisms controlling its activity, in the MG in homeostasis and during aging.

The MG is highly innervated[28–30], neurotransmitters can stimulate the proliferation of MG epithelial cells in vitro[52], and impaired innervation and/or neurotransmitter secretion have been observed during aging in other glands such as the exorbital lacrimal gland[62] and sweat gland[63]. However, the in vivo functions of MG innervation and how these might be altered in aging have not been examined. Our studies revealed that impaired innervation is a previously uncharacterized

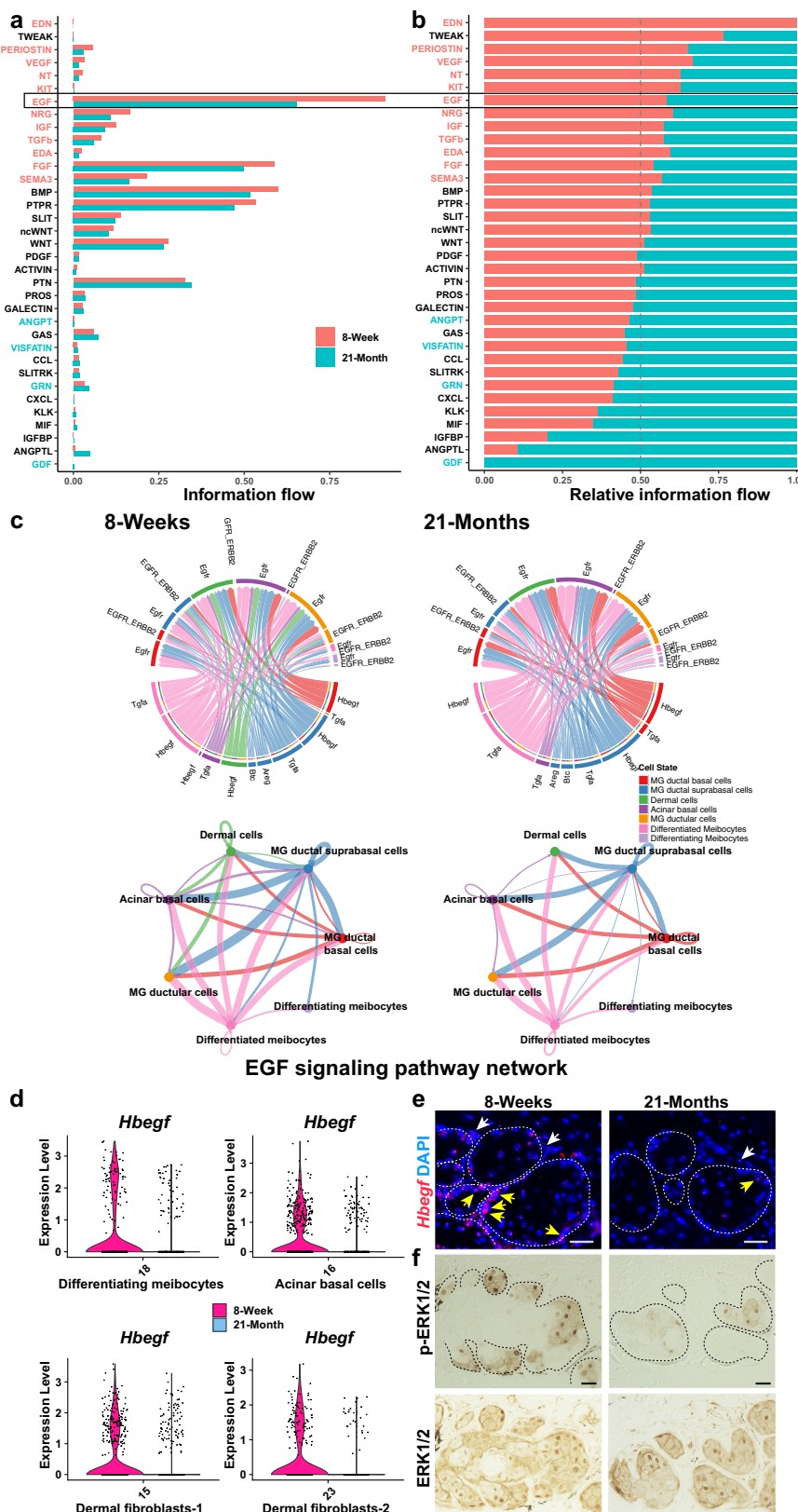

**Fig. 8 | HBEGF signaling is disrupted in aged MGs. a**, **b** CellChat analysis predicts decreased EGF signaling in aged MGs. **c** CellChat analysis predicts HBEGF signaling between dermal cells and MG ductular cells (green) at 8 weeks and its absence at 21 months. **d** Violin plots of snRNA-seq data show decreased percentages of *Hbegf*-expressing cells in the indicated cell populations in aged MG. **e** RNAscope shows reduced *Hbegf* expression levels per cell and fewer acinar basal cells (yellow arrows)

and surrounding dermal cells (white arrows) expressing *Hbegf* in aged MG. **f** IHC shows that p-ERK1/2 levels are decreased, but total ERK1/2 levels are similar in the acini of aged compared with young MG. White dashed lines in (**e**) and black dashed lines in (**f**) outline MG acini. Samples from *n* = 3 8-week and *n* = 3 21-month-old male C57BL/6 J mice were used for RNAscope and IHC. Scale bars represent 25 μm.

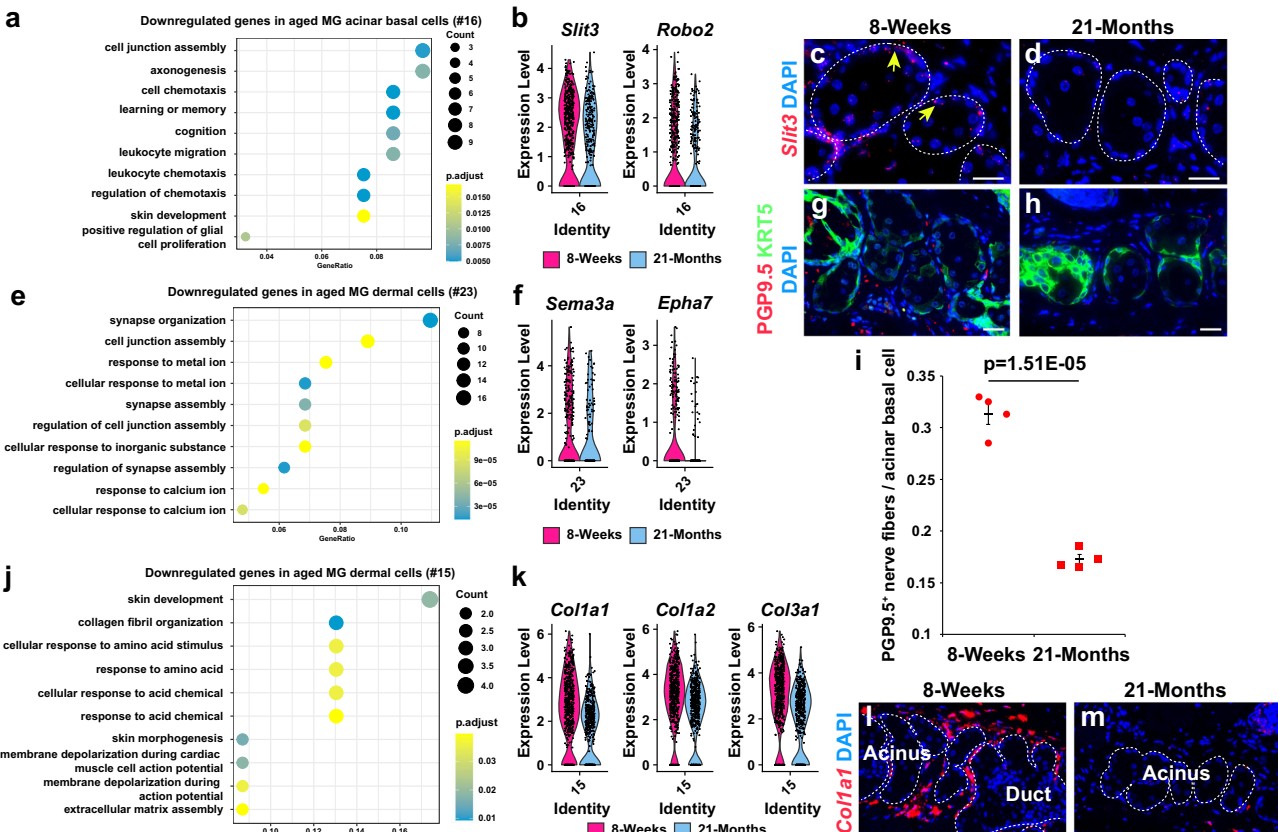

**Fig. 9 | Aged MGs exhibit reduced peripheral innervation and collagen I expression. a** GO analysis of genes with statistically significantly reduced expression in aged compared with young acinar basal cells (cluster #16). **b** Violin plots for neuronal guidance genes in acinar basal cells. **c, d)**Slit3 mRNA expression (yellow arrows) is reduced in aged MG acinar basal cells. **e, f** GO analysis of genes with reduced expression in aged cluster #23 dermal cells (**e**) and violin plots for neuronal guidance genes (**f**). **g, h** Reduced numbers of PGP9.5 nerve fibers (red) adjacent to the KRT5+ acinar basal layer (green) in aged mice. **i** Quantification of PGP9.5+ nerve fibers within 5 µM of each KRT5+ acinar basal cell in young and aged MGs. Independent biological samples from n = 4 8-week-old and *n* = 4 21-month-old male C57BL/6 J mice were used for quantification. At least 185 acinar basal cells were analyzed from each mouse. Significance in (**i**) was calculated by an unpaired two-tailed Student's t-test. Data are represented as mean +/− SEM. Source Data are provided as a Source Data file. **j, k** GO analysis of genes with reduced expression in aged cluster #15 dermal fibroblasts (**j**) and violin plots for type I collagen genes in dermal fibroblasts (**k**). **l, m** Reduced expression of *Col1a1* (red) in aged (**m**) versus young (**l**) dermis surrounding MG acini. For (**a, e, j**) FDR calculation was performed by clusterProfiler (v4.4.4) and EdgeR (v3.38.4) with the Benjamini-Hochberg procedure. Dot plots in (**a, e, j**) were generated with ggplot2. Violin plots in (**b, f, k**) were generated with the VlnPlot command from Seurat package. White dashed lines in (**c, d, l, m**) outline MG acini and/or ducts. Independent biological samples from *n* = 3 8-week and *n* = 3 21-month C57BL6/J mice were used for RNAscope, IF and IHC. Scale bars represent 25 µm. See also Supplementary Fig. S7.

feature of aged MG and may result from decreased expression of axon guidance factors in aged MG epithelial cells and surrounding dermis[64,65].

Limitations of the current study include that snRNA-seq, spatial transcriptomics, and aging-related experiments were carried out only in male mice. While most validation experiments were performed on both male and female samples, lack of inclusion of female samples in the snRNA-seq dataset and in aging-related experiments could mean that sex-specific differences in MG biology and aging were missed.

In summary, our data identify markers for distinct stem cell populations that maintain MG ducts and acini and reveal that MG ductules harbor stem cells for both the duct and the acinus. We uncovered Hh signaling as a key pathway regulating MG stem cell proliferation and showed that its activity is enhanced in human MGC and decreased in aging. We further found that decreased HBEGF signaling, reduced peripheral innervation, and an altered dermal microenvironment accompany MG aging and may contribute to reduced epithelial proliferation and MG dropout (Fig. 10). These observations provide an improved understanding of the mechanisms of MG homeostasis, aging, and tumorigenesis, and suggest Hh signaling as a therapeutic target in MGC and in MG aging. Transient, localized, and tightly controlled Hh and EGFR pathway activation could provide therapeutic benefit for patients

with evaporative dry eye disease by stimulating stem cell activity, whereas inhibition of the Hh pathway, for instance with SMO inhibitors[66], may be useful in the treatment of MGC.

## Methods

### Mice

C57BL/6 J mice (Jackson Laboratory #000664) were used for snRNA-seq, validation of snRNA-seq data via RNAscope and IF, spatial transcriptomics, and aging-related experiments. All other mice were maintained on a mixed C57BL/6 J / SJL/J FVB/NJ strain background. Up to five mice were maintained per cage in a specific pathogen-free barrier facility on a 12-hour light/12-hour dark cycle at 65-75°F with 40-60% humidity and were fed standard rodent laboratory chow. Mice of both sexes were analyzed in all experiments except for snRNA-seq, spatial transcriptomics, and aging-related experiments which used only male mice. No notable differences were noted in data from male and female mice. The following mouse lines were utilized: *Lrig1-Cre^ERT2* (Jackson Laboratory, #018418), *Lgr6-Cre^ERT2* (Jackson Laboratory, #016934), *Axin2-Cre^ERT2*[33], *Slc1a3-Cre^ERT2* (European Mouse Mutant Archive, EM:12216), *Gli2-Cre^ERT2*[67], *Krt5-rtTA* (Jackson Laboratory, #017519), *tetO-GLI2ΔN*[38], *Rosa26^mTmG* (Jackson Laboratory, #007676), *Rosa26^nTnG* (Jackson Laboratory, #023537), *KRT14:H2BGFP*[68], *KRT14-*

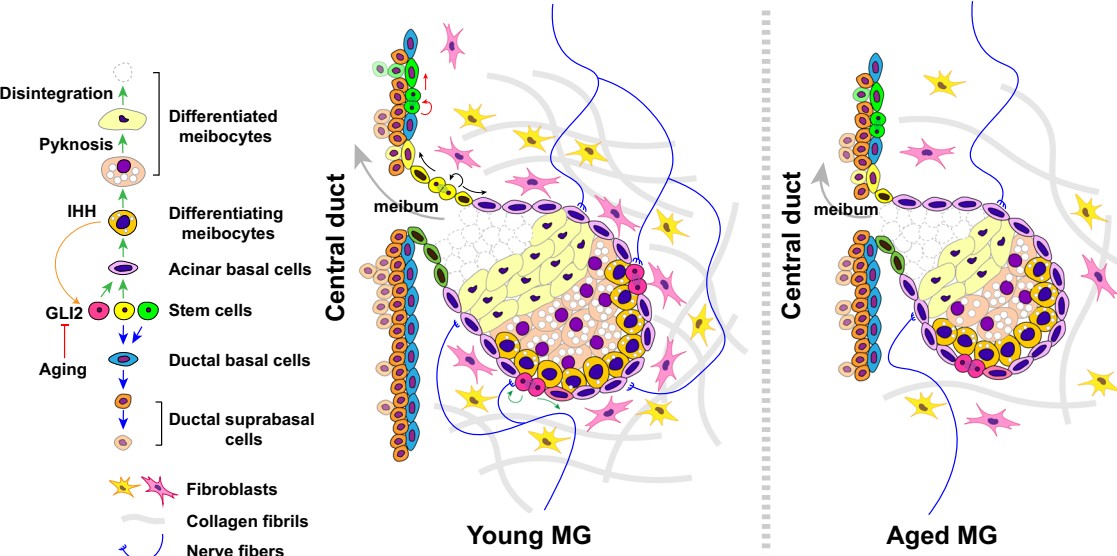

**Fig. 10 | Model for the effects of aging in the MG.** Hh signaling is reduced in aging due to inhibited GLI2 transcriptional activity, resulting in lower levels of stem cell proliferation. In parallel, peripheral innervation and collagen fibril density are reduced in aged compared with young MGs.

*Cre^{ERT2}* (Jackson Laboratory, #005107), *KRT14-Cre^{ER}* (Jackson Laboratory #005107), *Smo^{fl/fl}* (Jackson Laboratory, #004526), *Hdac1^{fl/fl} Hdac2^{fl/fl}*[69], *tetO-Cre* (Jackson Laboratory #006234), *Ptch1^{fl/fl}* (Jackson Laboratory #012457). All animal experiments were carried out under approved animal protocols according to institutional guidelines established by the Icahn School of Medicine at Mount Sinai IACUC, the University of Michigan IACUC, the University of Pennsylvania IACUC, or the Johns Hopkins University IACUC. Experiments with *KRT14-Cre^{ERT2} Smo^{fl/fl}* mice were carried out under Johns Hopkins University protocol #MO22M104, PI Carlo Iomini; Experiments with *KRT14-Cre^{ERT} Ptch1^{fl/fl}* mice were carried out under University of Michigan Unit for Laboratory Animal Medicine protocol #PRO00003165, PI Sunny Wong; experiments with *K5-rtTA tetO-GLI2ΔN* mice in Fig. 5a–c were carried out under University of Pennsylvania IACUC protocol #804809, PI, Sarah E. Millar; all other experiments with mice were carried out under Icahn School of Medicine at Mount Sinai IACUC protocol #2019-0028, PI Sarah E. Millar.

## Human MG samples

Human eyelid tissues were provided by the Department of Ophthalmology at the University of Pennsylvania as de-identified samples that had been discarded from unrelated surgical procedures. Use of the human tissue samples in research was approved by the University of Pennsylvania IRB under protocol #827926, PI Vivian Lee. Human eyelid samples were paraffin-embedded and sectioned for histological analysis, RNAscope, and IF.

## snRNA-seq

Eight 8-week-old and eight 21-month-old male mice were euthanized, and their eyelids were cut with scissors. The dorsal portion of the eyelids and the surrounding tissues near the tarsal plates were removed using a scalpel. The remaining tarsal plates containing MGs were digested with 0.2% collagenase A (Sigma-Aldrich #10103578001) in PBS for 1 hour at 37 °C. The tarsal plates were then rinsed twice with PBS, and tissues surrounding the MGs were removed with fine-tip forceps. Tissues from four littermate mice were pooled together in PBS for each replicate. The MG tissue-containing solutions were then transferred into 1.5 mL centrifuge tubes using Pasteur pipettes. The tissue solution was centrifuged at 300 g/min for 5 minutes, the supernatant was carefully removed, and the tissue pellet was rinsed once with PBS. Isolation of nuclei was performed using the protocol

(CG000375 Rev B) from 10x Genomics. RNA-seq libraries were prepared based on the 10x Genomics protocol (CG000315 Rev E) using 10000 nuclei per replicate. Single-cell separation was performed using the Chromium Next GEM Chip G Single Cell Kit (10x Genomics #PN-1000120). The RNA-seq libraries were sequenced in paired-end mode on an Illumina Novaseq platform with the Novaseq S4 flowcell.

## Alignment of 10x Genomics data

Cell Ranger Single-Cell Software Suite (v7.0.1, 10x Genomics) was used to align and quantify sequencing data against the mm10 mouse reference genome (refdata-gex-mm10-2020-A).

## Analysis of snRNA-seq data

Cell clustering and differential gene expression were analyzed using Seurat[70] (v4.3.0). For each replicate, high-quality nuclei (defined as nFeature_RNA > 200 & nFeature_RNA < 4500 & nCount_RNA > 500 & <nCount_RNA < 25000 & precent.mt <5) were analyzed. 6676 nuclei were analyzed from 8-Week replicate 1, 10734 from 8-Week replicate 2, 8336 from 21-Month replicate 1, and 7305 from 21-Month replicate 2. The CellCycleScoring function of Seurat was applied to mitigate the effects of cell cycle heterogeneity in the datasets. The SCTransform function of Seurat was used to integrate the four datasets. In total, 34 clusters were identified with a resolution of 1. For Violin plots, the y-axis was plotted against the log normalized data from the data slot of the Seurat object. Seurat employs a global-scaling normalization method "LogNormalize" that normalizes the feature expression (each gene) measurements for each cell by the total expression, multiplies this by a scale factor (10,000 by default), and log-transforms the result.

For RNA velocity and pseudotime analyses, bam files generated from Cell Ranger were processed using Velocyto (v0.6)'s run10x function with default parameters, resulting in loom files that contained counts of unspliced and spliced mRNAs. The loom files were merged with the h5ad files, which covered the meibomian gland clusters. Subsequently, the merged files were preprocessed with scVelo (v0.3)[71], and RNA velocity and pseudotime analyses were performed using cellDancer (v1.1)[72].

To identify DEGs, genes encoding ribosomal proteins were initially defined and removed by grepping for "^Rp(sl)" from the datasets before pseudobulk analysis using EdgeR (v3.38.4)[73]. Genes with an FDR (padj)<0.05 and |Log₂FC|>0.5 were considered DEGs and were subjected to gene ontology analysis with clusterProfiler (v4.4.4)[74]. FDR

calculation was performed by the EdgeR or clusterProfiler using the Benjamini-Hochberg procedure.

The interactions among distinct MG cell types (MG ductal basal cells, ductal suprabasal cells, ductular cells, acinar basal cells, differentiating meibocytes, and differentiated meibocytes), along with surrounding dermal cells (two populations were combined in this context), were analyzed using CellChat (v1.6.1)[43]. Briefly, the normalized counts were loaded into CellChat and preprocessed with default parameters. To establish the ligand-receptor interaction database, we selected secreted signaling pathways and utilized the "projectData" function to project gene expression data onto the mouse protein-protein interaction (PPI). To visualize specific signaling pathways, we employed the setting "type=truncatedMean, trim = 0.1" to compute the communication probability (computeCommunProb) except for Hh signaling (trim =0.001 was used). Core CellChat functions such as "computeCommunProbPathway", "aggregateNet", "netAnalysis_signalingRole", and "identifyCommunicationPatterns" were applied with standard parameters for downstream analysis.

To compare gene expression between Meibomian and sebaceous glands, MG acinar basal cells from our dataset and SG basal cells from SG RNA-seq dataset GSE225252[16] were isolated according to cell type annotations with Seurat's subset function. Individual basal cell datasets for Meibomian and sebaceous glands were further merged and processed for Violin plotting in Seurat (v4.3.0).

## Spatial transcriptomics
The 10x Visium Spatial Gene Expression kit (10x Genomics # PN-1000184) was employed to construct the spatial transcriptomic library, according to the manufacturer's instructions. A pair of upper and lower eyelids from an 8-week-old male mouse was dissected, sectioned at 10 μM, and mounted onto the assay slide containing four capture areas (6.5 mm×6.5 mm) with spatially barcoded poly T capture probes. After permeabilization, the captured mRNAs underwent reverse transcription, followed by cleaving from the slide. cDNA amplification, fragmentation, end-repair, poly A-tailing, adapter ligation, and sample indexing were performed in accordance with the manufacturer's protocol. The resulting cDNA libraries were quantified using TapeStation (Agilent) and Qubit (Invitrogen) and then subjected to sequencing in paired-end mode on a NovaSeq instrument (Illumina) at a depth of 50,000-100,000 reads per capture spot. The sequencing data were aligned and quantified using Space Ranger Software Suite (v2.1.0, 10x Genomics) against the reference genome.

## Analysis of spatial transcriptomic data
Integration of snRNA-seq data and spatial transcriptomic data was conducted through CytoSPACE (v1.0.5)[75]. Briefly, a non-normalized count-based gene expression file and a cell-type label file were generated from the snRNA-seq datasets. Additionally, a compressed spatial transcriptomes input file for CytoSPACE was created using files from Space Ranger outputs. Subsequently, all the input files were loaded into CytoSPACE with the recommended parameter settings, as detailed on the project's GitHub repository (https://github.com/digitalcytometry/cytospace).

## Immunofluorescence (IF) and immunohistochemistry (IHC)
Paraffin sections were deparaffinized and rehydrated to PBS. Antigen retrieval was performed with antigen unmasking solution (Vector Laboratories # H-3300). Sections were washed with PBST (PBS + 0.1% Tween 20) and blocked with 1% BSA for 30 minutes. Primary antibodies were applied, and the sections were incubated overnight at 4 °C. For IF, sections were subjected to three washes of PBST, followed by 1-hour incubation with secondary antibodies. The following antibodies were used: Rat monoclonal anti-Ki-67 (eBioscience #13-5689-82; 1:200); Rabbit anti-COL17A1 (Abcam #ab184996, 1:500, RRID AB_3073438); Rabbit anti-KRT17 (Abcam # ab109725, 1:200, RRID AB_10889888);

Mouse anti-KRT10 (Novus #NBP2-47650, 1:200); Rabbit anti-LOR (BioLegend #905103, 1:500, RRID AB_2734676); Rabbit anti-PPARγ (Cell Signaling Technology #2435 T, 1:200, RRID AB_2166051); Rabbit anti-FASN (Cell Signaling Technology #3180 T, 1:200, RRID AB_2100796); Guinea Pig anti-PLIN2 (Fitzgerald #20R-AP002, 1:500, AB_1282475); Rabbit anti-PLIN2 (Sigma-Aldrich #393A-1, 1:100); Rabbit anti-GFP (Abcam #ab290, 1:1000, RRID AB_2313768); Rabbit anti-KRT5 (BioLegend #905504, 1:500, RRID AB_2616956); Rabbit anti-KRT14 (Thermo Fisher Scientific #MA5-11599, 1:400, RRID AB_10982092); Rabbit anti-ERK1/2 (Cell Signaling Technology #4695 T, 1:100, RRID AB_390779); Rabbit anti-p-ERK1/2 (Cell Signaling Technology #4370 T, 1:100, RRID AB_2315112); Mouse anti-Myctag (Cell Signaling Technology #2276S, 1:1000, RRID AB_331783); Rabbit anti-GLI2 (Novus Biologicals #NB600-874SS, 1:200, RRID AB_10001953); Goat anti-GLI2 (R&D Systems #AF-3635, 1:50, RRID AB_2111902); Rabbit anti-Acetylated-Lysine (Cell Signaling Technology #9441 s, 1:100, RRID AB_331805); Rabbit anti-HDAC1 (Thermo Fisher Scientific #49-1025, 1:1000, RRID AB_2533875); Rabbit anti-HDAC2 (Thermo Fisher Scientific #51-5100, 1:1000, RRID AB_2533908); Donkey anti-Rabbit IgG Secondary Antibody, Alexa Fluor 555 (Invitrogen #A-31572, 1:800, RRID AB_162543); Donkey anti-Mouse IgG Secondary Antibody, Alexa Fluor 488 (Invitrogen #A-21202, 1:800, RRID AB_162543); Goat anti-Guinea Pig, Biotinylated (Vector Laboratories #A-7000, 1:500, RRID AB_2336132); Goat anti-Rabbit, Biotinylated (Vector Laboratories #BA-1000, 1:500, RRID AB_2313606); Streptavidin Fluorescein (Vector Laboratories #SA-5001, 1:500, RRID AB_2336462); Streptavidin Texas Red (Vector Laboratories #SA-5006, 1:500, RRID AB_2336754). The sections were washed with PBST and mounted with DAPI-containing medium (Invitrogen # P36930). For IHC, sections were washed with PBST and then incubated with 3% $H_2O_2$/PBST for 15 minutes, followed by two washes of PBST and 30-minute incubation with biotinylated secondary antibodies. Subsequent steps and DAB staining followed the manufacturer's instructions. The sections were dehydrated, cleared, and mounted. IF and IHC data were documented using a Leica Microsystems DM5500B microscope equipped with two cameras for fluorescent and bright field images (Leica Microsystems).

## RNAscope in situ hybridization
RNAscope was performed on paraffin sections using the RNAscope Multiplex Fluorescent Detection Kit v2 (Advanced Cell Diagnostics #323110), following the user's guide provided by the manufacturer. Probes were all obtained from Advanced Cell Diagnostics. The probes used were as follows: Mm-*Slc1a3* #430781; Mm-*Scd4* #486071; Mm-*Lrig1* #310521-C2; Mm-*Lgr6* 404961; Mm-*Axin2* #400331-C2; Mm-*Gli1* #311001-C2; Mm-*Ccnd1* #442671; Mm-*Smo* #318411-C2; Mm-*Gli2* #405771; Mm-*Shh* #314361-C2; Mm-Ihh #1259141-C1; Mm-*Ptch1* #402811; Mm-pan-*Hh* #415051; Mm-*Slit3* #542771; Mm-*Hbegf* #437601; Mm-*Col1a1* #319371; Hs-*GLI1* #310991-C2; Hs-*LRIG1* #407421; Hs-*LGR6* #410461.

## Tamoxifen induction
200 μl Tamoxifen (Sigma-Aldrich #T5648) dissolved in corn oil (10 mg/ml) was injected intraperitoneally once for lineage tracing. For gene deletion, daily intraperitoneal injections were performed on five consecutive days.

## Ex vivo live imaging of tarsal plate explants
Mice were euthanized, and the eyelids were dissected with scissors. Tissues surrounding the tarsal plates were removed using a surgical scalpel. The tarsal plates were placed in a 35-mm Lumox-bottom dish (Sarstedt #94.6077.331), immobilized with a custom-built holder, and incubated in DMEM/F12(1:1) medium (Gibco #21041-025) supplemented with 5% FBS for 2 hours in an environmental chamber supplied with 5% $CO_2$ at 37 °C before imaging. A Leica SP8 laser scanning confocal microscope equipped with Power HyD detectors and an

environmental chamber was used to image the cultured explants. Images were captured as z stacks at 3 μm intervals and tissues were imaged every 15 min with low laser power (<3%) with a HC PL APO CS2 10x/0.40 DRY objective lens. Stacked images were maximum projected with Leica LAS X 3.6.0.20104 software to generate the time-lapse video and video annotations were performed with Fiji software 2.1.0.

## Doxycycline administration

Doxycycline at 200 μg/ml (Sigma-Aldrich # D9891) together with 5% sucrose was fed to the mice continuously in the drinking water after initiation of its administration. The solution was freshly prepared every 5 days for induction periods of over 5 days.

## Bulk RNA-seq and data analysis

3 pairs of 6-week-old $Gli2\Delta N^{KrtSrtTA}$ mice and littermate controls were fed with 200 μg/ml doxycycline water for 4 days. Mice were euthanized and the upper and lower eyelids were dissected, frozen sectioned at 12 μm, and mounted onto polyethylene naphthalate-membrane slides (Leica Microsystems). The MGs from 25 sections from each sample were laser dissected using the gravity-assisted LMD 7000 system (Leica Microsystems) and collected by gravity assistance onto AdhensiveCap (Zeiss #415190-9211-000). RNA was extracted using the Qiagen All Prep DNA/RNA FFPE Kit (Qiagen #80234) according to the manufacturer's instructions. 50 ng RNA from each laser-captured sample was used for RNA-seq library construction using the Trio RNA-Seq™ library preparation kit (Nugene #0506). The libraries were sequenced on an Illumina HiSeq 4000 platform with the 100 bp single read (SR100) method. For data analysis, Salmon (v0.9.1)[76] was used to count data against the transcriptome defined in Gencode vM18. We then annotated and summarized the transcriptome count data to the gene level with tximeta[77] and performed further annotation with bio-maRt (v0.7.0)[78]. Normalizations and statistical analyses were carried out with DESeq2 (v1.20.0)[79]. Genes with padj<0.001 and |Log$_2$FC|>0.5 were selected for GO enrichment analysis, which was carried out by clusterProfiler (4.4.4). The results were plotted by ggplot2 (v3.4.0) and volcano plots were generated using EnhancedVolcano (v1.14.0).

## Proximity ligation assay (PLA)

PLA was carried out as previously described[33] using the DuoLink In Situ PLA kits (Sigma Aldrich #DUO92002 / #DUO92003 / #DUO92005). Briefly, 5 μm paraffin sections were dewaxed and rehydrated, followed by blocking with 5% donkey serum/PBS/0.8% Triton X-100 for 30 minutes and incubation with solutions containing primary antibodies overnight at 4 °C. The sections were washed with PBST and incubated in PLA probe anti-rabbit plus or anti-goat plus, and PLA probe anti-rabbit minus for 2 hours. The subsequent ligation and amplification steps followed the manufacturer's instructions. The sections were washed twice in washing buffer and once in PBS for 20 minutes each before mounting and imaging using a Leica SP8 laser scanning confocal microscope.

## Statistical analyses

Sample size for experiments involving quantitation was predetermined using the statsmodels (0.9.0) package in Python (3.7). Statistical analysis and graphical representation were performed using Microsoft Excel 2023. Unpaired or paired two-tailed Student's $t$-test was used to calculate statistical significance between two groups of data. $P < 0.05$ was considered significant. Data were represented as mean +/− SEM.

## Reporting summary

Further information on research design is available in the Nature Portfolio Reporting Summary linked to this article.

## Data availability

The authors declare that the main data supporting the findings of this study are available within the article and its Supplemental Information files. This study did not generate unique new reagents. snRNA-seq data generated in this study have been deposited in the GEO database under accession code GSE274498 "snRNA-seq of murine tarsal plates". The spatial transcriptomics data generated in this study have been deposited in the GEO database under accession code GSE274497 "Spatial transcriptomics of murine eyelid". Bulk RNA-seq data generated in this study have been deposited in the GEO database under accession code GSE274496 "Bulk RNA-sequencing of Meibomian glands from $Krt5$-$rtTA$ $tetO$-$GLI2\Delta N$ mice and littermate controls". Further information and requests for resources and reagents or for the minimum dataset necessary to interpret, verify, and extend the research in the article, should be directed to and will be fulfilled by Sarah E. Millar (sarah.millar@mssm.edu). No restrictions are placed on access to the minimum dataset. Source data are provided with this paper.

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

## Acknowledgements

We thank the Department of Ophthalmology at the University of Pennsylvania, Philadelphia, PA for de-identified discarded human eyelid tissue; Dr. Robert Sebra and the Genomics Core Facility, Icahn School of Medicine at Mount Sinai, New York, USA for snRNA sequencing and spatial transcriptomics; Dr. Steven Prouty for advice on laser capture; Drs. Elaine Fuchs and Michael Rendl for *KRT14:H2BGFP* mice; Dr. Adam Glick for *Krt5-rtTA* mice; and Dr. Eric N. Olson for *Hdac1^{fl/fl}* and *Hdac2^{fl/fl}* mice. This work was supported by NIH grants R01AR081322 (SEM); R37AR047709 (SEM); R01EY035337 (CI); R01EY036135 (CI); R01AR065409 (SYW); and a pilot grant from the Mount Sinai Skin Biology and Diseases Resource-based Center (SBDRC) P30AR079200 (XZ). Core facilities used in this study were provided by the Mount Sinai SBDRC P30AR079200 (PI, Elena Ezhkova); the University of Michigan SBDRC P30AR075043 (PI, Johann Gudjonsson); the University of Michigan Cancer Center P30CA046592 (PI, Eric Fearon); and the University of Pennsylvania SBDRC P30AR069589 (PI, Elizabeth Grice).

## Author contributions

Conceptualization: X.Z., M.X., S.E.M.; Formal Analysis: X.Z., M.X.; Funding Acquisition: X.Z., S.Y.W., C.I., S.E.M.; Investigation: X.Z., M.X., C.P., Y.L., A.F., S.Y.W.; Methodology: X.Z., M.X.; Project Administration: S.E.M.; Resources: X.Z., V.L., J.T.S., C.I., T.P., E.E.M., A.A.D., J.M.C., S.Y.W., S.E.M.; Supervision: C.I., S.E.M.; Validation: X.Z., M.X.; Visualization: X.Z., M.X.; Writing - Original Draft Preparation: X.Z.; Writing - Review and Editing: X.Z., M.X., S.Y.W., C.I., S.E.M.

## Competing interests

The authors declare no competing interests.
