## [Transparent Peer Review file · Nature Communications]

Identification of Meibomian gland stem cell populations and mechanisms of aging

Corresponding Author: Dr Sarah Millar

Version 0:

Reviewer comments:

Reviewer #1

(Remarks to the Author)

The meibomian gland (MG) is a fascinating yet understudied tissue crucial for tear film function, which is essential for sight. The MG is frequently affected by aging and can undergo transformation into cancer. However, very little is known about the molecular mechanisms that underlie MG maintenance and pathology. In this study, the authors investigate the stem cell populations in the meibomian gland, their markers, and key signaling pathways that control MG stem cells under homeostasis, as well as in the contexts of aging and cancer. They performed single-cell RNA sequencing and spatial transcriptomics to uncover the repertoire of cell populations in the young and aged MG of mice. Furthermore, they conducted extensive lineage tracing experiments using five different Cre drivers, including ex vivo live imaging, to explore the MG stem cell populations and their lineage hierarchy.

The authors demonstrated that four different Cre drivers (Lrig1, Lgr6, Axin2, and Gli2) similarly labeled both ductal and acinar basal cells of the MG. In all cases, long-term clonal expansion was observed in both ductal and acinar compartments, suggesting that these four promoters mark ductal and acinar stem cell populations. It is difficult to tell whether these populations overlap, but it is likely the case. At least under ex vivo culture, Lrig1 ductal cells were shown to contribute to acinus, however, it is difficult to conclude on homeostasis from this experiment as the authors correctly conclude. Moreover, they report that Slc1a1 which is exclusively expressed by acinar cells, marks acinar self-renewing stem cells. Taken together, this part of the paper is very novel and should serve the field for future work.

Next, the authors identified the hedgehog (Hh) and EGF signaling pathways that are active in MG stem cells. They elegantly demonstrated that Hh pathway inhibition significantly reduces stem cell proliferation and MG size, whereas Hh overexpression promotes stem cell growth and prominent MG expansion. Interestingly, they found an increase in Hh and stem cell markers in MG carcinoma in humans, and a reduction in aging. Finally, they demonstrated that niche-specific genes, including those involved in EGF signaling and collagen production, are downregulated in aged MG. While the paper could stand alone without reporting these correlative, yet, very interesting signaling pathways, the authors' decision to include it opens an interesting research avenue for future work and will probably increase the citation of the paper.

Altogether, this is an extensive work that will likely generate significant interest in the fields of stem cells, developmental biology, and ophthalmology. The use of multiple genetic and sequencing tools is a strength of the paper. The data are clear, well-controlled, and the conclusions are precise and carefully drawn.

I only have the following minor comments:

1. Fig1: for the broader readership, I strongly suggest to add a scheme that illustrates the gland structure (e.g. PMID:24926024), and maybe consider annotating the stem cell populations and/or genes. Also, the data in fig1e-i would be more easily visualized if presented on larger canvas space, perhaps in a separate figure.
2. I suggest to revise and shorten the text in lines 195-223, since the outcome of the tracing with the 4 Cre drivers is similar. I also suggest to revise the sentence on the conclusion from this important experiment to make it more clear and precise. Regarding the Slc1a3-Cre, it labeled acinar basal, differentiated, ductules and not central duct. On day 90, authors report detection of Slc1a3 traced cells in acinar basal, differentiated but not central duct. Can you specify about long term labeling in the ductules?
3. Promoters for tracing of central duct or ductule only would be useful for addressing renewal trajectories in future studies. It will be relevant to discuss this limit and need for future work.

Reviewer #2

(Remarks to the Author)

Zhu et al. identified some meibomian gland stem cell markers and reported Hedgehog signaling a key regulator of stem cell proliferation. The authors also demonstrated that decreased Hedgehog and EGF signaling were displayed in aged Meibomian glands. The reviewer read the manuscript with great interest. There are some concerns.

1. Hh in the Page3 Line 44 should be Hedgehog.
2. The mechanism behind tumor genesis is very complicated. The authors showed the MG carcinoma samples, and these results contributed little to those conclusions.
3. Aged MGs display decreased Hh and EGF signaling. What if the signaling was enhanced in aged MGs? The authors should design experiments to clarify it.
4. Human tissues were used. Did the authors get approval from any Ethics Committee?

Reviewer #3

(Remarks to the Author)

In this manuscript, the authors identified stem cell markers in distinct regions of meibomian glands (MGs) and revealed that the Hh signaling pathway is crucial for maintaining stem cell homeostasis. The authors further report that Hh signaling is increased in human MGCs and observe that aged MGs exhibit reduction of both Hh and EGF signaling, deficient innervation, and loss of collagen I in the niche. The current work provides a comprehensive investigation of cell populations and stem cell markers in meibomian glands and proposes key factors for maintaining stem cell homeostasis, thereby addressing a knowledge gap in understanding of MG stem cells.

The study is well-designed and executed, and the conclusions are interesting and novel. However, the mechanisms of innervation in young and aged MGs are not well established, the methods are inadequately described, and some data seem to be overinterpreted or misinterpreted.

I have several concerns with the manuscript and data, as presented:

1. Fig.1G: Axin2-labeled cells express MG ductal suprabasal cell markers at 2 days, after 120 days, Axin2-expressing cells are observed in MG ductal basal cells. This appears to be inconsistent with the conclusion of Fig.1B, which states that “the ductal basal cells give rise to ductal suprabasal cells” (line 156). How do the authors explain this result?

Fig.1E: There are fewer Lrig1+ positive cells in MGs, and their distribution seems to more closely align with the distribution of stem cells. Is it reasonable to use Lrig1 as a broad basal cell marker?

Fig.1H: The orange arrowhead appears to point to acinar basal cells rather than ductules. The note for the arrow is incorrect in lines 963-964.

2. Fig.2F-G: It is difficult to distinguish the expression changes of FANC from IHC result. The expression of FANC appears to be reduced in aged MGs. Please provide IF staining to further confirm this result.

3. Fig.4B, J, and L: Why is there no KRT14 staining signal in the acinar cells of MGCs?
What is the expression status of stem cell markers and Hh components in the ductal/ductular cells of MGCs.

4. Fig.5H-I: The authors mention that the down-regulation of Hh signaling in aged MGs is due to increased acetylation of Gli2, which mitigates its transcriptional activity. However, the authors also claim in Fig.5D-E that there are fewer cells expressing Gli2 in aged MGs compared to young MGs, especially in the ductules (line 354-355). Since acetylation modification of Gli2 alters its transcriptional activity but not its protein stability, how do the authors explain the decrease of Gli2+ cells in aged MGs?

Fig.5H-I: The PLA assay shows an increased association of Gli2 and acetyl-lysine in aged MGs. The PLA signal can only be detected when both Gli2 and acetyl-lysine are present. If there are fewer Gli2+ cells in aged MGs, why is the PLA signal in aged MGs much higher than that in young MGs?

Fig.5J-K: K5-rtTA tetO-Cre Hdac1fl/fl Hdac2fl/fl (Hdac1/2DcKO) only deletes Hdac1 and Hdac2 in acinar basal cells. Why do the authors observe increased PLA signals in differentiating acinar cells?

The outline of MGs in Fig.5J and K is difficult to distinguish.

Fig.5O-P: Why do the authors not detect the expression of HDAC1 and HDAC2 separately? Fig.5N shows that the expression changes of HDAC1 and HDAC2 in aged MGs are different. Co-staining of HDAC1 and HDAC2 may mask the actual changes in their expression in aged MGs.

5. Fig.6F: There are fewer acini observed in both young and aged MGs. Please provide clear figures to support this conclusion.

6. Extended Fig.1J: I do not observe any signal for Slca3 in this figure. Please provide clear figures to support the conclusion.
7. Extended Fig.3: The authors claim that “Lrig1+ cells in the ductule can contribute to the acinar basal layer.” However, RNAscope and lineage tracking indicate that Lrig1+ cells are present in both MG ducts (Fig. 1E, yellow arrows) and acinar basal cells. How do the authors reach the conclusion that Lrig1+ cells in acinar basal cells originate from ductular cells?
8. Both Gli2 and Gli3 are expressed in ductal and acinar cells according to the snRNA-seq result (Extended Fig.5A). However, the authors only examined the role of expressed Gli2 Δ N in acinar basal cells. What about the function of expressed Gli3 in acinar basal cells? Moreover, Gli2 and Gli3 demonstrate higher expression levels in ductal cells compared to acinar cells. Why do the authors not investigate the role of expressed Gli2 or Gli3 in ductal cells?
9. Hh signaling is a key regulator of maintaining proliferation in acinar basal stem cells. What is its function in ductal stem cells? More evidence is required to confirm the role of Hh signaling in maintaining stem cell homeostasis in MGs.
10. How does Gli2 Δ N affect acinar basal cell proliferation? Does it increase the proliferation rate or frequency?
11. In aged MGs, Hh signaling is reduced. What is the expression level of stem cell markers in aged MGs? Please provide more evidence.
12. The authors examined the expression pattern of Shh in Extended Fig.5B, which is located in acinar basal cells. In Extended Fig.6F, the authors show that Ihh is secreted from ductal basal cells, ductal suprabasal cells, and differentiating meibocytes. I would like to know which ligand of the Hh signaling is the main ligand in MG tissue for regulating stem cell proliferation, as Shh and Ihh play distinct roles in different tissues and developmental stages. Moreover, which ligand does “Pan-Hh” represent in Fig.5F-G?
13. Line 150-151: The statement “A key difference was that MG acinar basal cells and ductular cells expressed genes associated with neural guidance, including Slit3, Robo2, Sema3a, and EphA7” requires direct evidence or appropriate references.
14. Line 192: The statement “in line with the expression of endogenous Lrig1” requires additional information on the expression pattern of endogenous Lrig1.
15. Please provide detailed methods of the lineage tracing experiments, the catalog numbers for all primary antibodies used, and the probe sequences of the genes utilized for RNAscope in the method section.
16. In discussion section, the authors briefly summarize the results of the study but do not adequately discuss the article's highlights, unresolved questions, and advantages or disadvantages of this study compared to work of others. The authors should revise the discussion to include these aspects further.
17. The orientation of most figures in the manuscript has been rotated, please adjust the orientation of the figures to facilitate reading.
18. Line 80-82: The authors state, “Recent evidence suggests that the MG duct and acinus are separately maintained by distinct unipotent KRT14+ stem cells; however, KRT14 is ubiquitously expressed in the MG.” Please provide the relevant reference for this conclusion.
19. Line 91-96: Why do the authors refer to results from SGs instead of directly exploring potential new mechanisms from the snRNA-seq in maintaining MG stem cell homeostasis? Although two tissues show functional similarities, there are distinct differences in their structure and developmental processes.
20. Line 99: The authors explain the reason for using snRNA-seq instead of sc-RNA-seq. Since mouse MGs are not difficult to digest, scRNA-seq can provide more information compared to the snRNA-seq.
21. In the presented snRNA-seq data (Fig.1A), KRT17 was used to represent ductal suprabasal cells and ductular cells. However, in IF staining data (Extended Fig.1F), it is unclear whether KRT17 is also expressed in ductular cells. If so, what are the differences in expression level among these cell types?
Line 134: What is the rationale for using moderate expression of Krt17 and Lrig1 as annotations for the ductular cells, and what is the co-staining result of two makers in ductular cells.
22. The previous study demonstrated that DNase2 is highly expressed in meibocytes. What about its expression in cluster 18? Why do authors not use the DNase2 as a marker to label cluster 18?

Reviewer #4

(Remarks to the Author)

Zhu et al. report an elegant study on lineage tracing of the MG system. Using multiple lineage tracer lines in vivo, using drivers linked to stem cells in other contexts, they delineate molecular markers and possible pathways involved in stem cell

dynamics. The authors then focus on Hh signaling, using loss of Smo and gain of an artificial form of Gli2 to see diminished and enhanced epithelial lineage expansions, respectively. This involves Hh signaling in aspects of stem cell function in this system. Finally, the authors delve in more peripheral aspects to the core study, such as Gli2 and acetylated lysine, HBEGF signaling and neurite outgrowth guiding signals in aging.

Overall, this is an elegant and sound study regarding lineage tracing in vivo, with careful analyses of snRNAseq and microscopy. Whereas the study confirms and expands previous findings (Parfitt et al.), it provides important relationships (maps) of stem to derived differentiated cells in different parts of the MG. However, it does not completely clarify the role of Hh signaling in this system.

-Is there autocrine or paracrine Hh signaling, given that ligands and responders are found both in epithelial and surrounding stroma/mesenchyme.

-Are Gli2+ cells all Hh responsive or have they already responded? This is important as Gli2 has been shown to respond to multiple other signaling inputs.

-What are the cells that selectively express Gli1 (Fig. 4)? What is the fate of these cells?

-Are the effects of Gli2N'delta the same as loss of endogenous Ptch1?

-Is Gli2N'delta epistatic over Smo loss in all affected cells?

-Does Gli2N'delta have 'off-target' gain-of-function effects?

-Is Hh signaling sufficient for tumorigenesis in this system? The correlation with human tumors is good, but it is not proven mechanistically.

-The authors may want to draw on a possible parallel previously established between Hh and EGF functions, which may help to explain partial effects (reduced proliferation) of Smo deletion in acinar basal cells.

Thus, whereas the first part is outstanding methodologically and in a descriptive manner, several important questions remain on the more novel mechanistic part, namely the role of Hh signaling in this system. Further work tightening the experimentally deduced role of Hh signaling may make the paper appropriate for this journal.

Reviewer #5

(Remarks to the Author)

Version 1:

Reviewer comments:

Reviewer #1

(Remarks to the Author)

The authors have significantly improved the manuscript and I have no further suggestions.

Reviewer #2

(Remarks to the Author)

The authors answered all the questions raised by the reviewer. The quality of this manuscript has greatly improved.

Reviewer #3

(Remarks to the Author)

The authors have addressed my questions well. However, I still have one concern about Figure 4. The authors examined the effect of HH signaling on ductal proliferation by forced expression of GLI2ΔN in GLI2ΔN Krt5rtTA transgenic mice. However, since Krt5 is specifically expressed in the acinar basal cells, it is necessary to clarify or examine the expression region of GLI2ΔN in the MG duct. Additionally, what are the effects of knockdown of Ptc on ductal basal cells? Is the effect same as forced expression of GLI2ΔN?

Reviewer #4

(Remarks to the Author)

The revised paper provides a more concise and tight discussion of the main findings while expanding on critical questions raised by the referees. One key point raised was on the specificity of Gli2deltaN. The new data on Ptch1 loss of function is a critical addition. It would have been nice to follow this with RNAseq, as done with Gli2deltaN, to make sure the latter acts in a physiological manner regarding targets. This said, the revisions make this study, and mostly the first part, an important addition to the literature on MG research that will likely be a key reference in the future. I am thus inclined to favor acceptance after addition of the RNAseq comparison between Ptch1 KO and Gli2deltaN expression in basal cells, which could be done quickly.

Response to Reviewers

NCOMMS-24-54085-T: "Identification of Meibomian gland stem cell populations and mechanisms of aging" by Xuming Zhu et al.

We thank the reviewers for their thoughtful comments and excellent suggestions. We have fully responded to all of these, including providing new experimental data, as outlined below. We believe that these revisions have substantially increased the significance and impact of this study.

Please note that revisions are highlighted in yellow in the main text, figure legends, and supplementary figure legends.

Reviewer #1 - comments to the author

The meibomian gland (MG) is a fascinating yet understudied tissue crucial for tear film function, which is essential for sight. The MG is frequently affected by aging and can undergo transformation into cancer. However, very little is known about the molecular mechanisms that underlie MG maintenance and pathology. In this study, the authors investigate the stem cell populations in the meibomian gland, their markers, and key signaling pathways that control MG stem cells under homeostasis, as well as in the contexts of aging and cancer. They performed single-cell RNA sequencing and spatial transcriptomics to uncover the repertoire of cell populations in the young and aged MG of mice. Furthermore, they conducted extensive lineage tracing experiments using five different Cre drivers, including ex vivo live imaging, to explore the MG stem cell populations and their lineage hierarchy.

The authors demonstrated that four different Cre drivers (Lrig1, Lgr6, Axin2, and Gli2) similarly labeled both ductal and acinar basal cells of the MG. In all cases, long-term clonal expansion was observed in both ductal and acinar compartments, suggesting that these four promoters mark ductal and acinar stem cell populations. It is difficult to tell whether these populations overlap, but it is likely the case. At least under ex vivo culture, Lrig1 ductal cells were shown to contribute to acinus, however, it is difficult to conclude on homeostasis from this experiment as the authors correctly conclude. Moreover, they report that Slc1a1 which is exclusively expressed by acinar cells, marks acinar self-renewing stem cells. Taken together, this part of the paper is very novel and should serve the field for future work.

Next, the authors identified the hedgehog (Hh) and EGF signaling pathways that are active in MG stem cells. They elegantly demonstrated that Hh pathway inhibition significantly reduces stem cell proliferation and MG size, whereas Hh overexpression promotes stem cell growth and prominent MG expansion. Interestingly, they found an increase in Hh and stem cell markers in MG carcinoma in humans, and a reduction in aging. Finally, they demonstrated that niche-specific genes, including those involved in EGF signaling and collagen production, are downregulated in aged MG. While the paper could stand alone without reporting these correlative, yet, very interesting signaling pathways, the authors' decision to include it opens an interesting research avenue for future work and will probably increase the citation of the paper. Altogether, this is an extensive work that will likely generate significant interest in the fields of stem cells, developmental biology, and ophthalmology. The use of multiple genetic and sequencing tools is a strength of the paper. The data are clear, well-controlled, and the conclusions are precise and carefully drawn.

Response: We thank the reviewer for their positive comments on the novelty, significance, and rigor of this study.

I only have the following minor comments:

1. Fig1: for the broader readership, I strongly suggest to add a scheme that illustrates the gland structure (e.g. PMID:24926024), and maybe consider annotating the stem cell populations and/or genes. Also, the data in fig1e-i would be more easily visualized if presented on larger canvas space, perhaps in a separate figure.

Response: Thank you for these excellent suggestions. As recommended, we have now provided a scheme illustrating the MG structure with the stem cell populations annotated with marker genes (new Fig. 1b). This scheme is described in the revised text lines 157-158. Additionally, we have divided old Fig. 1 into two separate figures (new Fig. 1a-d and new Fig. 2a-f) in the revised manuscript, making it easier to visualize the data.

2. I suggest to revise and shorten the text in lines 195-223, since the outcome of the tracing with the 4 Cre drivers is similar. I also suggest to revise the sentence on the conclusion from this important experiment to make it more clear and precise. Regarding the *Slc1a3-Cre*, it labeled acinar basal, differentiated, ductules and not central duct. On day 90, authors report detection of *Slc1a3* traced cells in acinar basal, differentiated but not central duct. Can you specify about long term labeling in the ductules?

Response: Thank you for your excellent suggestion to shorten the text describing lineage tracing. We have followed this suggestion – please see lines 205-212.

We apologize for the mis-statement in the original text that *Slc1a3-Cre^{ERT2}* labels ductular cells after 2 days of lineage tracing. We did not in fact detect *Slc1a3-Cre^{ERT2}*-mediated labeling of the ductule at either 2 days or 90 days, despite detection of low levels of *Slc1a3* mRNA expression in ductules. This has now been corrected in the text lines 209-212:

“*Slc1a3-Cre^{ERT2}*-labeled cells were identified in the acinar basal layer (Fig. 2f, white arrowhead) and meibocytes (Fig. 2f, yellow arrowhead) but not in ductules or central duct. Absence of *Slc1a3-Cre^{ERT2}* labeling in ductules may be due to low *Slc1a3* promoter activity in ductule cells.”

We also revised the Discussion to more accurately summarize the lineage tracing data (lines 491-496 and lines 504-506):

“We found that basal stem cells marked by *Slc1a3-Cre^{ERT2}* exclusively contribute to the acinus; by contrast, lineage tracing analyses revealed that both duct and acinus were replenished by stem cells marked by *Lrig1-Cre^{ERT2}*, *Lgr6-Cre^{ERT2}*, *Axin2-Cre^{ERT2}* or *Gli2-Cre^{ERT2}*. Further studies will be needed to identify specific markers for stem cell populations that exclusively renew MG ductules and ducts.”

While *Slc1a3-Cre^{ERT2}* did not label ductules, likely due to low levels of *Slc1a3* promoter activity in ductule cells, lineage tracing combined with live imaging of MG explants showed that LRIG1+ cells in the ductule can migrate towards the acinus.

3. Promoters for tracing of central duct or ductule only would be useful for addressing renewal trajectories in future studies. It will be relevant to discuss this limit and need for future work.

Response: We agree with the reviewer. We have added the following sentence to the Discussion section of the revised manuscript (lines 494-496):

“Further studies will be needed to identify specific markers for stem cell populations that exclusively renew MG ductules and ducts”.

Reviewer #2 - comments to the author

Zhu et al. identified some meibomian gland stem cell markers and reported Hedgehog signaling a key regulator of stem cell proliferation. The authors also demonstrated that decreased Hedgehog and EGF signaling were displayed in aged Meibomian glands. The reviewer read the manuscript with great interest. There are some concerns.

Response: We thank the reviewer for their interest in this study.

1. *Hh in the Page3 Line 44 should be Hedgehog.*

Response: We thank the reviewer for noticing this and have corrected it accordingly (page 3 line 46 in revised text).

2. *The mechanism behind tumor genesis is very complicated. The authors showed the MG carcinoma samples, and these results contributed little to those conclusions.*

Response: We thank the reviewer for this point. We completely agree with the reviewer that the mechanism of MGC tumorigenesis is complex. We believe that our data on normal human MGs and human MGC provide a valuable addition to this study because:

- (i) The data show similar expression patterns for putative stem cell markers *GLI2*, *GLI1*, and *LRIG1* in human versus mouse MG, providing support for the mouse MG as a valid model for understanding human MG homeostasis.
- (ii) Our data showing expansion of stem cell markers in human MGC provide novel evidence that MGC involves expansion of MG stem cells.
- (iii) Our novel data revealing upregulated/expanded activity of Hh signaling pathway components in human MGC show that Hh signaling correlates with proliferation in human MG as well as in our mouse models.
- (iv) Our finding of increased Hh signaling in human MGC identifies a potential new therapeutic target.

In the revised manuscript, we now clarify that hyperactivation of Hh signaling via forced expression of *GLI2ΔN* or deletion of endogenous epithelial *Ptch1* in mouse MG results in increased basal cell proliferation and expansion, but not histological MGC, within the time frames of our studies. These data suggest that activated Hh signaling contributes to MGC proliferation, but may not be sufficient to initiate tumorigenesis. We believe that these observations provide a valuable contribution to the understanding of MGC etiology, which is currently poorly characterized. Please see lines 517-523 in revised text:

“To date, genetic mutations of Hh components have not been identified in MGC, suggesting that increased Hh signaling, while contributing to MGC proliferation, may not be a primary driver of

MGC. In line with this, forced expression of *GLI2ΔN* or deletion of epithelial *Ptch1* resulted in stem cell proliferation and basal cell expansion in our mouse models, but was not sufficient to produce histological indications of MGC⁵⁸ in the time frames of our studies. Further investigation is required to delineate the mechanisms underlying MGC and the contributions of Hh signaling to this disease.”

3. Aged MGs display decreased Hh and EGF signaling. What if the signaling was enhanced in aged MGs? The authors should design experiments to clarify it.

Response: We agree with the reviewer that these are interesting experiments. However, exploring the potential of Hh and EGF activators in pre-clinical aging models is a major endeavor that is beyond the scope of the current study and will be the subject of future work.

4. Human tissues were used. Did the authors get approval from any Ethics Committee?

Response: The human tissue samples used in this study were provided to the authors by the Department of Ophthalmology at the University of Pennsylvania and the Institutional Biorepository & Pathology Core at the Icahn School of Medicine at Mount Sinai as de-identified samples that had been discarded from unrelated clinical procedures. These studies are therefore not considered as human subjects research, and do not require the authors to have IRB approval. This has been clarified in the revised text (lines 605-609):

“Human eyelid tissues were provided by the Department of Ophthalmology at the University of Pennsylvania and the Institutional Biorepository & Pathology Core at the Icahn School of Medicine at Mount Sinai as de-identified samples that had been discarded from unrelated surgical procedures.”

Reviewer #3 - comments to the Author

In this manuscript, the authors identified stem cell markers in distinct regions of meibomian glands (MGs) and revealed that the Hh signaling pathway is crucial for maintaining stem cell homeostasis. The authors further report that Hh signaling is increased in human MGCs and observe that aged MGs exhibit reduction of both Hh and EGF signaling, deficient innervation, and loss of collagen I in the niche. The current work provides a comprehensive investigation of cell populations and stem cell markers in meibomian glands and proposes key factors for maintaining stem cell homeostasis, thereby addressing a knowledge gap in understanding of MG stem cells.

The study is well-designed and executed, and the conclusions are interesting and novel. However, the mechanisms of innervation in young and aged MGs are not well established, the methods are inadequately described, and some data seem to be overinterpreted or misinterpreted.

Response: We thank the reviewer for their positive comments on the manuscript and their suggestions for improvements.

I have several concerns with the manuscript and data, as presented:

1. Fig.1G: Axin2-labeled cells express MG ductal suprabasal cell markers at 2 days, after 120 days, Axin2-expressing cells are observed in MG ductal basal cells. This appears to be inconsistent with the conclusion of Fig.1B, which states that “the ductal basal cells give rise to

ductal suprabasal cells” (line 156). How do the authors explain this result?

Response: *Axin2* labels basal cells as well as suprabasal cells in the duct at 2 days. We have replaced the original figure with a more representative image that shows this more clearly (new Fig. 2d). While we cannot exclude the potential existence of stem cells in the ductal suprabasal layer, the results of lineage tracing analysis with *Axin2-Cre^{ERT2}* and the absence of proliferation in ductal suprabasal cells (Supplementary Fig. S1e) support the prediction from Velocity analysis that ductal basal cells give rise to ductal suprabasal cells.

2. Fig.1E: There are fewer Lrig1+ positive cells in MGs, and their distribution seems to more closely align with the distribution of stem cells. Is it reasonable to use Lrig1 as a broad basal cell marker?

Response: Our data do not identify *Lrig1* as a broad basal cell marker as it is also highly expressed in duct suprabasal cells (please see revised Supplementary Fig. 1m-o and Supplementary Fig. 2b,d,e).

3. Fig.1H: The orange arrowhead appears to point to acinar basal cells rather than ductules. The note for the arrow is incorrect in lines 963-964.

Response: We identified ductules in tissue sections as lying at the junction between the central duct and the acinus. The orange arrowheads in original Fig. 1H / new Fig. 2e indicate this location. This is clarified in the revised text (lines 188-190):

“Ductular cells (cluster 17) were identified in sectioned tissue as lying at the junction of the acinus and the duct.”

4. Fig.2F-G: It is difficult to distinguish the expression changes of FANC from IHC result. The expression of FANC appears to be reduced in aged MGs. Please provide IF staining to further confirm this result.

Response: Unfortunately, the FASN antibody did not work for IF despite repeated attempts. We have replaced old Fig. 2G / new Fig. 3g with a more representative image.

5. Fig.4B, J, and L: Why is there no KRT14 staining signal in the acinar cells of MGCs?

Response: We thank the reviewer for raising this interesting point. We speculate that loss of KRT14 expression may reflect epithelial-mesenchymal transition of the tumor cells. We have now highlighted the loss of KRT14 expression in central cells of human MGC and discussed a possible reason for this in the revised text (lines 339-341):

“Central cells in the MGC samples exhibited downregulation of KRT14 (Fig. 6b, j, l) compared with control MGs (Fig. 6a, i, k); loss of KRT14 expression may reflect epithelial-mesenchymal transition of the tumor cells.”

6. What is the expression status of stem cell markers and Hh components in the ductal/ductular cells of MGCs.

Response: We thank the reviewer for this question. In the revised text we now note that the MGC samples showed loss of an organized structure, and MG ducts and ductules were not apparent in the tumor area. Please see the revised text lines 338-339:

“The MGC samples showed loss of an organized structure, and MG ducts and ductules were not apparent in the tumor area (Fig. 6a-l).”

7. Fig.5H-I: The authors mention that the down-regulation of Hh signaling in aged MGs is due to increased acetylation of Gli2, which mitigates its transcriptional activity. However, the authors also claim in Fig.5D-E that there are fewer cells expressing Gli2 in aged MGs compared to young MGs, especially in the ductules (line 354-355). Since acetylation modification of Gli2 alters its transcriptional activity but not its protein stability, how do the authors explain the decrease of Gli2+ cells in aged MGs?

Response: We thank the reviewer for this interesting question. There are two possible explanations for the decrease in the numbers of cells expressing *Gli2* mRNA in aged MGs. Firstly, as expression of *Gli2* mRNA is itself regulated by Hh signaling (e.g. reference #44), increased acetylation of GLI2 protein could result in decreased expression of its own mRNA; alternatively, or in addition, *Gli2* mRNA expression may mark a specific stem cell population that is decreased in aging. We now clarify this in the Results section lines 378-387 and 396-399:

“Validation by RNAscope revealed that there were fewer cells expressing *Ptch1* mRNA in the acinar basal layer, differentiating meibocyte population, and stroma in aged MG (Fig. 7c) compared with MG at 8 weeks (Fig. 7b). Additionally, there were fewer cells expressing *Gli2* mRNA in aged compared with young MG, especially in the ductules (Fig. 7d, e, white arrows). By contrast, expression of the stem cell markers *Lrig1*, *Lgr6*, *Slc1a3* and *Axin2* in ductular or acinar basal cells did not differ appreciably in aged compared with young MGs (Supplementary Fig. S6j, k). These data may suggest that a specific *Gli2*+ stem cell population in ductules is reduced in aging; alternatively, as Hh signaling can upregulate expression of *Gli2* as well as *Ptch1* mRNA⁴⁴, the reduced number of *Gli2*+ cells may reflect decreased signaling through the Hh pathway.”

“Acetylation mitigates the transcriptional activity of GLI2⁴⁵; thus, this mechanism could potentially account for the observed decrease in the percentages of cells expressing Hh-regulated genes, including *Ptch1* and *Gli2* itself, in MGs and their stromal environment.”

We also discuss this result in the Discussion section lines 525-531:

“Interestingly, although MG basal cell proliferation was reduced in aging, expression of the stem cell markers *Lrig1*, *Lgr6*, *Slc1a3* and *Axin2* was not appreciably altered in aged versus young MG ductular and acinar basal cells. By contrast, aged versus young MGs had fewer cells expressing *Gli2*, which we found also marks MG stem cells. As *Gli2* can be upregulated by Hh signaling⁴⁴, the decline in *Gli2* mRNA expression may reflect the decrease in Hh pathway activity that we observed in aged MGs. Alternatively, it is possible that a specific *Gli2*+ stem cell population is reduced in aging. Further work will be needed to distinguish these mechanisms.”

8. Fig.5H-I: The PLA assay shows an increased association of Gli2 and acetyl-lysine in aged MGs. The PLA signal can only be detected when both Gli2 and acetyl-lysine are present. If there are fewer Gli2+ cells in aged MGs, why is the PLA signal in aged MGs much higher than that in young MGs?

Response: Although the relative percentage of cells expressing *Gli2* mRNA was reduced in aged MGs, within those cells that still expressed GLI2 protein in aged MG and surrounding

stroma, its acetylation was increased. In other words, the level of GLI2 protein acetylation per expressing cell was increased. We now clarify this in the Results section, lines 392-396:

“While fewer cells expressing *Gli2* mRNA were present in aged compared with young MGs (Fig. 7d, e), proximity ligation assays (PLA) revealed that subsets of the cells that still expressed GLI2 protein in aged MG and surrounding stroma displayed its elevated association with acetylated lysine, both within the acini (Fig. 7h, i, white arrows) and in the surrounding stroma (Fig. 7h, i, blue arrows).”

We also highlight this point in the Discussion, lines 534-538:

“Expression levels of Hh ligands were similar in young and aged MGs; however, while there were fewer cells expressing *Gli2* mRNA in aged MGs, those cells with remaining GLI2 protein displayed increased association of GLI2 with acetylated lysine, which dampens its transcriptional activity, and could result in decreased expression of Hh-regulated genes including *Gli2* itself.”

9. Fig.5J-K: *K5-rtTA tetO-Cre Hdac1^{fl/fl} Hdac2^{fl/fl} (Hdac1/2DcKO)* only deletes *Hdac1* and *Hdac2* in acinar basal cells. Why do the authors observe increased PLA signals in differentiating acinar cells?

Response: We induced *Hdac1/2* deletion in MG basal cells of *Krt5-rtTA tetO-Cre Hdac1^{fl/fl} Hdac2^{fl/fl}* mice; as K5+ acinar and ductal stem cells give rise to differentiating acinar and duct cells, this approach results in *Hdac1/2* deletion in the entire epithelial MG within one 9-day cycle of renewal. Please see reference #46. This has now been clarified in the Results text lines 404-407:

“To test this, we induced *Hdac1/2* deletion in KRT5+ MG basal cells of *Krt5-rtTA tetO-Cre Hdac1^{fl/fl} Hdac2^{fl/fl}* mice; as KRT5+ acinar and ductal stem cells give rise to differentiating acinar and duct cells, this approach results in *Hdac1/2* deletion in the entire epithelial MG within 9 days of induction⁴⁶.”

The corresponding Figure is revised Fig. 7j-m.

10. The outline of MGs in Fig.5J and K is difficult to distinguish.

Response: We have replaced these images with clearer ones – please see revised Fig. 7j, k.

11. Fig.5O-P: Why do the authors not detect the expression of HDAC1 and HDAC2 separately? Fig.5N shows that the expression changes of HDAC1 and HDAC2 in aged MGs are different. Co-staining of HDAC1 and HDAC2 may mask the actual changes in their expression in aged MGs.

Response: We agree with the reviewer that original Fig. 5N / revised Fig. 7n shows changes in the expression levels of *Hdac1* and *Hdac2* mRNAs in certain subsets of cells in aged versus young MGs. As complete deletion of *Hdac1* or *Hdac2* alone does not impact MG morphology (please see reference #46), these changes are unlikely to account for age-related MG phenotypes. This is now explained in the revised text, lines 417-418:

“As deletion of *Hdac1* or *Hdac2* alone does not impact MG morphology⁴⁶, these changes are unlikely to account for age-related MG phenotypes.”

To address the reviewer's question about levels of individual HDAC1 and HDAC2 proteins in aged versus young MGs, we have now performed separate IF experiments with antibodies specific for HDAC1 or HDAC2. Please see new Fig. 7o-r. These results are described in the revised text lines 418-420:

"Furthermore, IF assays failed to detect an appreciable difference in the expression of HDAC1 or HDAC2 proteins between young and aged MG acini (Fig. 7o-r)."

12. Fig.6F: There are fewer acini observed in both young and aged MGs. Please provide clear figures to support this conclusion.

Response: We have provided more representative examples of pERK1/2 and ERK1/2 IHC staining for 21-months MG samples in revised Fig. 8f.

13. Extended Fig.1J: I do not observe any signal for *Slcla3* in this figure. Please provide clear figures to support the conclusion.

Response: We have replaced the image for *Slcla3* RNAscope with an image that more clearly shows the ISH signal. Please see revised Supplementary Fig. S1j.

14. Extended Fig.3: The authors claim that "*Lrig1*+ cells in the ductule can contribute to the acinar basal layer." However, RNAscope and lineage tracking indicate that *Lrig1*+ cells are present in both MG ducts (Fig. 1E, yellow arrows) and acinar basal cells. How do the authors reach the conclusion that *Lrig1*+ cells in acinar basal cells originate from ductular cells?

Response: Our conclusion that *Lrig1*+ ductular cells can contribute to the acinar basal layer is based on the ex vivo imaging data with MG explants from *Lrig1-Cre^{ERT2} Rosa26^{nTnG}* mice (Supplementary Figs S3, S4 and Supplementary Videos S1 and S2). We now clarify that, based on in vivo lineage tracing data, acinar basal cells also self-renew. Please see revised text lines 250-252:

"Based on lineage tracing data (Fig. 2), acinar basal stem cells also self-renew, and this is likely the major mechanism for their replenishment."

15. Both *Gli2* and *Gli3* are expressed in ductal and acinar cells according to the snRNA-seq result (Extended Fig.5A). However, the authors only examined the role of expressed *Gli2ΔN* in acinar basal cells. What about the function of expressed *Gli3* in acinar basal cells? Moreover, *Gli2* and *Gli3* demonstrate higher expression levels in ductal cells compared to acinar cells. Why do the authors not investigate the role of expressed *Gli2* or *Gli3* in ductal cells?

Response: We thank the author for these questions. In response, we have now analyzed proliferation in MG duct basal cells expressing *GLI2ΔN* versus controls and show that proliferation of duct basal cells is statistically significantly increased by forced *GLI2ΔN* expression (new Fig. 4o, p, r); revised text lines 280-281; 289-291:

"Hyper-activation of Hh signaling in MG epithelium causes expansion of acinar and ductal basal cells"

"Similarly, forced expression of *GLI2ΔN*+ caused a significant increase in basal cell proliferation in MG ducts (Fig. 4o, p, r)."

Also, relevant to this question, we now show that deletion of epithelial *Smo* causes statistically significantly decreased proliferation of duct basal cells (Fig. 3j, k, m); revised text lines 276-278:

“*Smo*-deleted MGs exhibited significantly decreased proliferation of acinar and ductal basal cells compared with controls (Fig. 3h-m). Thus, Hh signaling promotes the proliferation of both acinar and ductal basal cells.”

Gli3 generally represses Hh pathway target gene expression. Its deletion might be predicted to produce a phenotype similar to that of *Smo* knockout, and it is certainly possible that it plays additional or other roles in the MG. This is an interesting question but is beyond the scope of the current study.

16. *Hh* signaling is a key regulator of maintaining proliferation in acinar basal stem cells. What is its function in ductal stem cells? More evidence is required to confirm the role of Hh signaling in maintaining stem cell homeostasis in MGs.

Response: We have now assayed for the proliferation of ductal basal cells in *Gli2ΔN*-expressing and *Smo*-deficient MGs and have incorporated the data in revised Figs. 3 and 4. Please see the response to point #15 for details.

17. How does *Gli2ΔN* affect acinar basal cell proliferation? Does it increase the proliferation rate or frequency?

Response: We quantified the percentage of Ki-67+ cells, which represents the proliferation frequency. We have modified the text accordingly. Please see revised text lines 287-291:

“After 7 days, most meibocytes had been replaced by *GLI2ΔN*+ cells (Fig. 4e, f, k, i) that showed statistically significantly increased proliferation frequency compared to control acinar basal cells (Fig. 4m, n, q). Similarly, forced expression of *GLI2ΔN*+ caused a significant increase in basal cell proliferation frequency in MG ducts (Fig. 4o, p, r).”

18. In aged MGs, Hh signaling is reduced. What is the expression level of stem cell markers in aged MGs? Please provide more evidence.

Response: Expression of the stem cell marker *Gli2* in aged versus young MGs was presented in the original manuscript and is shown in revised Fig. 7a, d, e. In response to the reviewer’s question, we have now analyzed expression of *Lrig1*, *Lgr6*, *Slc1a3* and *Axin2* in snRNA-seq data from aged versus young MGs (please see revised Supplementary Fig. S6j) and validated the data by RNAscope ISH (revised Supplementary Fig. S6k).

Interestingly, while there were fewer cells expressing *Gli2* in the MG and surrounding stroma of aged versus young MGs, the relative expression levels of the other stem cell markers were unchanged in aging. These data may suggest that a specific *Gli2*+ stem cell population is reduced in aging; alternatively, as Hh signaling can upregulate expression of *Gli2* as well as *Ptch1* mRNA (e.g. reference #44), the reduced number of *Gli2*+ cells may reflect decreased signaling through the Hh pathway. These results are discussed in the revised Results section lines 378-387:

“Validation by RNAscope revealed that there were fewer cells expressing *Ptch1* mRNA in the acinar basal layer, differentiating meibocyte population, and stroma in aged MG (Fig. 7c)

compared with MG at 8 weeks (Fig. 7b). Additionally, there were fewer cells expressing *Gli2* mRNA in aged compared with young MG, especially in the ductules (Fig. 7d, e, white arrows). By contrast, expression of the stem cell markers *Lrig1*, *Lgr6*, *Slc1a3* and *Axin2* in ductular or acinar basal cells did not differ appreciably in aged compared with young MGs (Supplementary Fig. S6j, k). These data may suggest that a specific *Gli2*⁺ stem cell population in ductules is reduced in aging; alternatively, as Hh signaling can upregulate expression of *Gli2* as well as *Ptch1* mRNA⁴⁴, the reduced number of *Gli2*⁺ cells may reflect decreased signaling through the Hh pathway.”

We also discuss this result in the Discussion section lines 525-531:

“Interestingly, although MG basal cell proliferation was reduced in aging, expression of the stem cell markers *Lrig1*, *Lgr6*, *Slc1a3* and *Axin2* was not appreciably altered in aged versus young MG ductular and acinar basal cells. By contrast, aged versus young MGs had fewer cells expressing *Gli2*, which we found also marks MG stem cells. As *Gli2* can be upregulated by Hh signaling⁴⁴, the decline in *Gli2* mRNA expression may reflect the decrease in Hh pathway activity that we observed in aged MGs. Alternatively, it is possible that a specific *Gli2*⁺ stem cell population is reduced in aging. Further work will be needed to distinguish these mechanisms.”

19. The authors examined the expression pattern of *Shh* in Extended Fig.5B, which is located in acinar basal cells. In Extended Fig.6F, the authors show that *Ihh* is secreted from ductal basal cells, ductal suprabasal cells, and differentiating meibocytes. I would like to know which ligand of the Hh signaling is the main ligand in MG tissue for regulating stem cell proliferation, as *Shh* and *Ihh* play distinct roles in different tissues and developmental stages. Moreover, which ligand does “Pan-Hh” represent in Fig.5F-G?

Response: In response to the reviewer’s question, we have now examined *Ihh* expression in the MG using RNAscope. Please see revised Supplementary Fig. S5c and revised text lines 260-263:

“*Ihh* and *Ptch1* localize to acinar basal cells (Supplementary Fig. S5c, d, green arrows), differentiating meibocytes (Supplementary Fig. S5c, d, white arrow), and ductular cells (Supplementary Fig. S5c, d, yellow arrows)”

The Pan-Hh RNAscope probe detects both *Shh* and *Ihh*; this is clarified in the revised Figure 7 figure legend.

CellChat analysis of the snRNA-seq data predicted that IHH is the major ligand of importance in MG homeostasis. Providing definitive evidence for which of these is the main ligand in MG tissue for regulating stem cell proliferation would require analysis of conditional *Shh* and *Ihh* mutant mice. This is an interesting question but is beyond the scope of the current study.

20. Line 150-151: The statement “A key difference was that MG acinar basal cells and ductular cells expressed genes associated with neural guidance, including *Slit3*, *Robo2*, *Sema3a*, and *Epha7*” requires direct evidence or appropriate references.

Response: We thank the reviewer for raising this important point. In response, we have carried out a more detailed comparison of gene expression between Meibomian and sebaceous glands. For this, MG acinar basal cells from our dataset and SG basal cells from SG RNA-seq dataset GSE225252 (reference #16) were isolated according to cell type annotations with Seurat’s subset function. Individual basal cell datasets for Meibomian and sebaceous glands were further

merged and processed for Violin plotting in Seurat (v4.3.0) (revised Supplementary Fig. S1r). The snRNA-seq data were validated by RNAscope (revised Supplementary Fig. S1s). This analysis allowed us to state more accurately that genes associated with neural guidance, such as *Slit3* and *Robo2*, were expressed at relatively higher levels in MG acinar basal and ductular cells compared with HF-associated SG basal cells. The methods are described in the Methods section, lines 671-675:

“To compare gene expression between Meibomian and sebaceous glands, MG acinar basal cells from our dataset and SG basal cells from SG RNA-seq dataset GSE225252¹⁶ were isolated according to cell type annotations with Seurat's subset function. Individual basal cell datasets for Meibomian and sebaceous glands were further merged and processed for Violin plotting in Seurat (v4.3.0).”

The results are described in the Results section, lines 162-165:

“A key difference was that genes associated with neural guidance, such as *Slit3* and *Robo2*, were expressed at relatively higher levels in MG acinar basal and ductular cells compared with HF-associated SG basal cells (Supplementary Fig. S1r, s).”

21. Line 192: The statement “in line with the expression of endogenous *Lrig1*” requires additional information on the expression pattern of endogenous *Lrig1*.

Response: We now provide additional analysis of the expression of endogenous *Lrig1* in snRNA-seq data in Supplementary Fig. S1m, n, showing low levels of *Lrig1* expression in acinar basal cells, and higher levels in ductules and duct. The data were validated by RNAscope ISH, shown in Supplementary Fig. S1o and Supplementary Fig. S2e. Please also see revised text lines 141-143:

“ductular cells (cluster 17) by moderate expression of *Krt17* and *Lrig1* (Supplementary Fig. S1m-o; Supplementary Fig. S2e), and low levels of *Slc1a3* and *Pparg* (Supplementary Fig. S1a, j);”

22. Please provide detailed methods of the lineage tracing experiments, the catalog numbers for all primary antibodies used, and the probe sequences of the genes utilized for RNAscope in the method section.

Response: Mouse lines used for lineage tracing are described in Methods lines 593-597. Tamoxifen induction for lineage tracing is described in Methods lines 747-749. The time frames for induction and analysis are described in the relevant sections of Results and are shown schematically in revised Fig. 2a and Fig. 5n-s. The precise alleles used in each experiment are described in Results and are indicated in revised Fig. 2b-f and Fig. 5n-s. Details of the antibodies used, including catalog number, dilution, and RRID information, have been moved from the original supplemental table to Methods lines 705-728. All the probes used for RNAscope are commercially available from Advanced Cell Diagnostics; the corresponding information including catalog number has been moved from the original supplemental table to Methods lines 740-745. The Accession Number, target region, and number of probe pairs generated to the target region for each probe are readily available on the Advanced Cell Diagnostics website.

23. In discussion section, the authors briefly summarize the results of the study but do not adequately discuss the article's highlights, unresolved questions, and advantages or

disadvantages of this study compared to work of others. The authors should revise the discussion to include these aspects further.

Response: We thank the reviewer for this suggestion. We have revised the Discussion section to more clearly point out the article's highlights and novelty compared with previous work; please see revised text lines 482-488. We also include more discussion of unresolved questions raised by our studies, and areas that require further investigation. Please see revised text lines 494-496; 517-523; 525-531; 551-556.

24. The orientation of most figures in the manuscript has been rotated, please adjust the orientation of the figures to facilitate reading.

Response: We have rearranged the panels in most of the figures so that they are now in a vertical orientation for easier reading; we did not alter the orientation of Fig. 9 because this resulted in the panels being too small to clearly display key information.

25. Line 80-82: The authors state, "Recent evidence suggests that the MG duct and acinus are separately maintained by distinct unipotent KRT14+ stem cells; however, KRT14 is ubiquitously expressed in the MG." Please provide the relevant reference for this conclusion.

Response: We now provide the relevant references (#12 and #17) as requested. Please see revised text lines 85-86:

"Recent evidence suggests that the MG duct and acinus are separately maintained by distinct unipotent KRT14+ stem cells¹²; however, KRT14 is ubiquitously expressed in the MG¹⁷."

26. Line 91-96: Why do the authors refer to results from SGs instead of directly exploring potential new mechanisms from the snRNA-seq in maintaining MG stem cell homeostasis? Although two tissues show functional similarities, there are distinct differences in their structure and developmental processes.

Response: We respectfully disagree with the reviewer's description of this work. In this study, we provide a comprehensive and unbiased view of MG homeostasis and aging through snRNA-seq, and Velocity and Pseudotime analyses, as well as using existing information about the development and maintenance of HF-associated SGs to inform genetic and lineage tracing experiments. While our study reveals some commonalities with the SG, we also reveal many novel aspects of the MG not previously described in SGs, including expression of *Gli2* in MG stem cells; higher levels of expression of neural guidance genes in MG versus SG; a role for ductule structures in harboring MG stem cells; decreased Hh and EGF signaling in aging; and loss of peripheral innervation and stromal collagen in aging.

27. Line 99: The authors explain the reason for using snRNA-seq instead of sc-RNA-seq. Since mouse MGs are not difficult to digest, scRNA-seq can provide more information compared to the snRNA-seq.

Response: We chose to employ snRNA-seq rather than scRNA-seq because lipid-rich differentiated MG acinar cells are fragile and cannot easily be collected by centrifugation. Indeed, this limitation is noted in recent publications that used scRNA-seq to analyze SGs (reference #16) and MGs (reference #25); by contrast nuclei from lipid-rich cells can be pelleted easily allowing us to capture the full range of MG cells including differentiated meibocytes. This is pointed out in the revised Introduction, lines 105-108:

“We chose to employ snRNA-seq rather than scRNA-seq because lipid-rich differentiated MG acinar cells are fragile and cannot easily be collected by centrifugation^{16, 25}; by contrast nuclei from lipid-rich cells can be pelleted easily allowing us to capture the full range of MG cells including differentiated meibocytes.”

28. In the presented snRNA-seq data (Fig. 1A), KRT17 was used to represent ductal suprabasal cells and ductular cells. However, in IF staining data (Extended Fig. 1F), it is unclear whether KRT17 is also expressed in ductular cells. If so, what are the differences in expression level among these cell types?

Response: To address this question, we provide more detailed analyses of snRNA-seq data in revised Supplementary Fig. S1m, n, showing that *Krt17* mRNA is most highly expressed in suprabasal duct cells (cluster 8) and is moderately expressed in ductular cells (cluster 17). IF analyses showed that KRT17 protein localizes to MG ducts (Supplementary Fig. S1o, white arrows) and ductules (Supplementary Fig. S1o, yellow arrows). Please see figure legend for Supplementary Fig. S1m-o and revised main text lines 138-143.

29. Line 134: What is the rationale for using moderate expression of *Krt17* and *Lrig1* as annotations for the ductular cells, and what is the co-staining result of two makers in ductular cells.

Response: To address this question, as noted above we provide more detailed analyses of snRNA-seq data in revised Supplementary Fig. S1m, n. The yellow color in the feature plots shown in Supplementary Fig. S1m indicates overlapping expression of *Krt17* and *Lrig1*. The dot plots in Supplementary Fig. S1n show that *Krt17* mRNA is most highly expressed in suprabasal duct cells (cluster 8) and is moderately expressed in ductular cells (cluster 17); *Lrig1* mRNA is moderately expressed in ductal suprabasal cells (cluster 8) and ductular cells (cluster 17). Combined IF for KRT17 protein and RNAscope for *Lrig1* mRNA (Supplementary Fig. S1o) showed their overlapping expression in MG ducts (white arrows) and ductules (yellow arrows). Please see figure legend for Supplementary Fig. S1m-o and revised main text lines 138-143.

30. The previous study demonstrated that *DNase2* is highly expressed in meibocytes. What about its expression in cluster 18? Why do authors not use the *DNase2* as a marker to label cluster 18?

Response: To address this question, we analyzed expression of *Dnase2a* in our snRNA-seq data. This analysis confirmed expression of *Dnase2a* in differentiated meibocytes (cluster 18) but showed that it is also moderately expressed in differentiating meibocytes (cluster 24) and detectably expressed in a number of other MG cell types including ductular cells (cluster 17) and acinar basal cells (cluster 16), as well as being moderately expressed in hair follicle matrix cells (cluster 19), and hair follicle outer root sheath cells (cluster 26). By comparison, *Scd4* is more specifically expressed in differentiated meibocytes, and *Scd3* localizes quite specifically to differentiating and differentiated meibocytes. Please see the data in the revised Supplementary Fig. 1p and the corresponding figure legend. We also discuss this in the revised Results text, lines 146-149:

“While *Dnase2* has been described as a marker for differentiated meibocytes in human MGs²⁶, we found that it is also expressed in other cell sub-populations in murine MGs (Supplementary Fig. S1p); this could be due to species-specific differences.”

Reviewer #4 - comments to the author

Zhu et al. report an elegant study on lineage tracing of the MG system. Using multiple lineage tracer lines in vivo, using drivers linked to stem cells in other contexts, they delineate molecular markers and possible pathways involved in stem cell dynamics. The authors then focus on Hh signaling, using loss of Smo and gain of an artificial form of Gli2 to see diminished and enhanced epithelial lineage expansions, respectively. This involves Hh signaling in aspects of stem cell function in this system. Finally, the authors delve in more peripheral aspects to the core study, such as Gli2 and acetylated lysine, HBEGF signaling and neurite outgrowth guiding signals in aging.

Overall, this is an elegant and sound study regarding lineage tracing in vivo, with careful analyses of snRNAseq and microscopy. Whereas the study confirms and expands previous findings (Parfitt et al.), it provides important relationships (maps) of stem to derived differentiated cells indifferent parts of the MG. However, it does not completely clarify the role of Hh signaling in this system.

Response: We thank the reviewer for their positive comments on our manuscript and their thoughtful questions.

1. Is there autocrine or paracrine Hh signaling, given that ligands and responders are found both in epithelial and surrounding stroma/mesenchyme.

Response: CellChat analysis of the snRNA-seq data predicted that IHH ligands secreted from MG ductal cells and differentiating Meibocytes are received by ductular cells, acinar basal cells, and dermal cells in a paracrine fashion. The effects of epithelial *Smo* deletion are consistent with this model but do not rule out autocrine signaling or provide information on the effects of reception of Hh ligands by dermal cells. We have added a comment to this effect in Results, lines 368-374:

“Analysis of snRNA-seq data from young and aged MGs using CellChat, which assesses intercellular communication based on patterns of ligand and cognate receptor expression⁴³, predicted that in young mice, IHH ligands secreted from MG ductal cells and differentiating Meibocytes are received by ductular cells, acinar basal cells, and dermal cells (Supplementary Fig. S6f, g). These analyses suggest that Hh ligands act in a paracrine fashion in young MGs, but do not exclude autocrine mechanisms.”

2. Are Gli2+ cells all Hh responsive or have they already responded? This is important as Gli2 has been shown to respond to multiple other signaling inputs.

Response: While Shh-dependent activation of *Gli2* is required for Shh functions in the skin (PMID: 12533516) and expression of *Gli2* is enhanced by Hh signaling (reference #44), our analyses do not distinguish whether all *Gli2*+ cells are Hh responsive. For this reason, we also analyzed the expression of the Hh target genes *Gli1* and *Ptch1* in our analyses of the MG. Additionally, in new experiments, we found that inducible deletion of epithelial *Ptch1* had similar effects of basal cell expansion to forced expression of *GLI2ΔN*, suggesting that *GLI2ΔN* acts predominantly by activating the Hh pathway – please see response to point #4 below. Conversely, epithelial *Smo* deletion decreased basal cell proliferation, providing further evidence for the pro-proliferative role of endogenous Hh signaling in MG basal cells.

3. What are the cells that selectively express Gli1 (Fig. 4)? What is the fate of these cells?

Response: In murine MG, *Gli1* mRNA localizes to acinar basal cells (Supplementary Fig. S5f, green arrows), differentiating meibocytes (Supplementary Fig. S5f, white arrow), and ductular cells (Supplementary Fig. S5f, yellow arrow). We also observed *GLI1* mRNA in acinar basal cells of normal human MG (Fig. 6c, yellow arrows). As our data indicate that acinar basal cells differentiate into meibocytes, and ductular cells can give rise to duct, it is likely that *Gli1*⁺ cells adopt these fates. Genetic lineage tracing experiments with a *Gli1* promoter-driven inducible Cre line would be required to provide definitive evidence for this; these would be interesting to perform but are beyond the scope of the current study.

4. Are the effects of *Gli2N*'delta the same as loss of endogenous *Ptch1*?

Response: Given that GLI2ΔN could have off-target effects, we agree with the reviewer that this is an important question. To address this, we have now examined the effects of inducible epithelial deletion of endogenous *Ptch1* in the MGs of adult *KRT14-Cre^{ERT} Ptch1^{fl/fl}* mice 12 weeks after induction. Consistent with the effects of forced expression of GLI2ΔN, we found that loss of epithelial *Ptch1* resulted in overgrowth of KRT5⁺ acinar basal cells (revised Fig. 4s-v). These data are described in Results, lines 293-300:

“As GLI2ΔN is not subject to the same regulation as endogenous GLI2, it might produce off-target effects that differ from those resulting from loss of endogenous Hh signaling. To control for this, we examined the effects of inducible epithelial deletion of the endogenous *Ptch1* Hh inhibitory receptor in the MGs of adult *KRT14-Cre^{ERT} Ptch1^{fl/fl}* mice 12 weeks after induction. Consistent with the effects of forced expression of GLI2ΔN, we found that loss of epithelial *Ptch1* resulted in overgrowth of KRT5⁺ acinar basal cells (Fig. 4s-v). Thus, hyper-activation of Hh signaling, either via forced expression of GLI2ΔN, or through deletion of endogenous epithelial *Ptch1*, caused basal cell expansion.”

Please also see Discussion, lines 511-514:

“Our snRNA-seq data and analyses of mice lacking the Hh receptor *Smo* revealed that Hh signaling plays a key role in promoting proliferation of MG stem cells. Consistent with this, forced expression of activated GLI2 or inducible deletion of *Ptch1* in MG epithelial cells resulted in MG basal cell expansion.”

5. Is *Gli2N*'delta epistatic over *Smo* loss in all affected cells?

Response: We thank the reviewer for this interesting question. As we found that epithelial deletion of endogenous *Ptch1* caused MG basal cell expansion, similar to the effects of forced GLI2ΔN expression, and as *Smo* and *Ptch1* are well established to function in the same pathway, it is likely that GLI2ΔN is epistatic over *Smo* loss in the MG. Demonstrating this definitively would require generation of mice carrying all 5 alleles (*KRT14-Cre^{ERT2}*, *Smo^{fl/fl}*, *Krt5-rtTA*, *tetO-GLI2ΔN*) and would likely take more than a year, which is not feasible in a reasonable time frame for revision of this paper. We are also not entirely convinced of the likely novelty and impact of the resulting data, given what we already know about the Hh pathway.

6. Does *Gli2N*'delta have 'off-target' gain-of-function effects?

Response: We agree with the reviewer that, given the GLI2 Δ N is not subject to the normal regulation of GLI2 protein, it could have off target effects. To control for this, we have now generated mice with inducible epithelial deletion of *Ptch1* and show that this also leads to expansion of MG basal cells. Please see the detailed response to question #4, above.

7. Is Hh signaling sufficient for tumorigenesis in this system? The correlation with human tumors is good, but it is not proven mechanistically.

Response: Our studies show that expression of stem cell markers is expanded, and Hh pathway components are upregulated, in MGC, suggesting that MGC involves expansion of MG stem cells and providing correlative evidence for the pro-proliferative functions of Hh signaling in the human MG. However, forced expression of GLI2 Δ N or deletion of epithelial *Ptch1* resulted in stem cell proliferation and basal cell expansion in our mouse models but was not sufficient to produce histological indications of MGC (reference #58) in the time frames of our studies. Further investigation is required to delineate the mechanisms underlying MGC and the contributions of Hh signaling to this disease but is beyond the scope of the current study. We now discuss this in the Discussion section, lines 517-523:

“To date, genetic mutations of Hh components have not been identified in MGC, suggesting that increased Hh signaling, while contributing to MGC proliferation, may not be a primary driver of MGC. In line with this, forced expression of GLI2 Δ N or deletion of epithelial *Ptch1* resulted in stem cell proliferation and basal cell expansion in our mouse models, but was not sufficient to produce histological indications of MGC⁵⁸ in the time frames of our studies. Further investigation is required to delineate the mechanisms underlying MGC and the contributions of Hh signaling to this disease.”

Please also see the response to Reviewer 2, point #2.

8. The authors may want to draw on a possible parallel previously established between Hh and EGF functions, which may help to explain partial effects (reduced proliferation) of Smo deletion in acinar basal cells.

Response: We thank the reviewer for this excellent suggestion. We have now included discussion of possible interactions of Hh and EGF signaling pathways in the MG in the Discussion section, lines 551-556:

“EGFR signaling displays extensive crosstalk with the Hh pathway in other cellular contexts; for instance, Shh modulates EGFR-dependent proliferation in embryonic stem cells, neural stem cells, and keratinocytes by transactivating EGFR^{59, 60, 61}. Further research will be needed to determine the precise role of HBEGF-EGFR signaling, its interactions with Hh signaling, and the mechanisms controlling its activity, in the MG in homeostasis and during aging.”

9. Thus, whereas the first part is outstanding methodologically and in a descriptive manner, several important questions remain on the more novel mechanistic part, namely the role of Hh signaling in this system. Further work tightening the experimentally deduced role of Hh signaling may make the paper appropriate for this journal.

Response: We thank the reviewer for their comments and questions. In response, we have provided extensive clarifications and new experimental data, including analysis of MG phenotypes in mice lacking epithelial *Ptch1*. We hope the reviewer agrees that these definitive

new genetic loss of function data enhance the significance and mechanistic aspects of this work.

Reviewer #5 - comments to the author

Response: Thank you for co-reviewing.

NCOMMS-24-54085-T: "Identification of Meibomian gland stem cell populations and mechanisms of aging" by Xuming Zhu et al.

Response to Reviewers

We thank the reviewers for their extremely thoughtful comments which guided us in substantially improving this manuscript.

Reviewer #1 (Remarks to the Author):

The authors have significantly improved the manuscript and i have no further suggestions

We thank the reviewer for their positive comments on our study.

Reviewer #2 (Remarks to the Author):

The authors answered all the questions raised by the reviewer. The quality of this manuscript has greatly improved.

We thank the reviewer for their positive comments on our study.

Reviewer #3 (Remarks to the Author):

The authors have addressed my questions well. However, I still have one concern about Figure4. The authors examined the effect of HH signaling on ductal proliferation by forced expression of GLI2ΔN in GLI2ΔN Krt5rtTA transgenic mice. However, since Krt5 is specifically expressed in the acinar basal cells, it is necessary to clarify or examine the expression region of GLI2ΔN in the MG duct.

We now clarify that the *Krt5* promoter drives expression in ductal basal cells as well as in acinar basal cells. Please see line 283 in the revised manuscript and immunofluorescence for KRT5 protein in duct as well as acinar basal cells in Figure 4g-l. This explains why *GLI2ΔN* is expressed in duct basal cells as well as in acinar basal cells in *K5-rtTA tetO-GLI2ΔN* mice (Figure 4p).

Additionally, what are the effects of knockdown of Ptc on ductal basal cells? Is the effect same as forced expression of GLI2ΔN?

Ptch1 deletion had less effect on the morphology of duct compared with acinar basal cells; we now note this in lines 298-299. We also now note that, while forced expression of *GLI2ΔN* significantly increased the proliferation of duct as well as acinar basal cells, it also had a less pronounced effect on duct compared with acinar morphology (lines 290-291).

Reviewer #4 (Remarks to the Author):

The revised paper provides a more concise and tight discussion of the main findings while expanding on critical questions raised by the referees. One key point raised was on the specificity of Gli2deltaN. The new data on Ptch1 loss of function is a critical addition. It would have been nice to follow this with RNAseq, as done with Gli2deltaN, to make sure the latter acts

*in a physiological manner regarding targets. This said, the revisions make this study, and mostly the first part, an important addition to the literature on MG research that will likely be a key reference in the future. I am thus inclined to favor acceptance after addition of the RNAseq comparison between *Ptch1* KO and *Gli2deltaN* expression in basal cells, which could be done quickly.*

We thank the reviewer for their positive comments on the manuscript. We agree that it would be interesting to generate RNA-seq data from *Ptch1* KO mice and compare the results with those from forced expression of *GLI2ΔN*. However, this experiment is beyond the scope of the current study and could not be completed in a reasonable time frame for revision of the manuscript.